# Low-frequency solar radio type II bursts and their association with space weather events during ascending phase of solar cycle 25

Theogene Ndacyayisenga[1,2], Jean Uwamahoro[3], Jean Claude Uwamahoro[1], Daniel Izuikedinachi Okoh[2,4], Kantepalli Sasikumar Raja[5], Akeem Babatunde Rabiu[2], Christian Kwisanga[1], and Christian Monstein[6]

[1]University of Rwanda - College of Science and Technology, Kigali, P.O.BOX 3900, Rwanda
[2]United Nations African Regional Centre for Space Science and Technology Education - English (UN-ARCSSTE-E), Obafemi Awolowo University Campus, Ile Ife, Nigeria
[3]University of Rwanda, College of Education, P.O. BOX 55, Rwamagana – Rwanda
[4]National Institute for Geophysics and Volcanology (INGV), 00143 Rome, Italy
[5]Indian Institute of Astrophysics, II Block, Koramangala, Bengaluru - 560 034, India
[6]IRSOL, Istituto Ricerche Solari "Aldo e Cele Daccò", Università della Svizzera italiana, Locarno, Switzerland.

**Correspondence:** Theogene Ndacyayisenga (ndacyatheogene@gmail.com)

**Abstract.** Type II solar radio bursts are signatures of the coronal shocks and, therefore, particle acceleration events in the solar atmosphere and interplanetary space. Type II bursts can serve as a proxy to provide early warnings of incoming solar storm disturbances, such as geomagnetic storms and radiation storms, which may further lead to ionospheric effects. In this article, we report the first observation of 32 Type II bursts by measuring various plasma parameters that occurred between May 2021 and December 2022 in solar cycle 25. We further evaluated their accompanying space weather events in terms of ionospheric total electron content (TEC) enhancement using the rate of TEC index (ROTI). In this study, we find that at heliocentric distance $\sim 1 - 2\ R\odot$, the shock and the Alfvén speeds are in the range 504 - 1282 $\mathrm{kms^{-1}}$ and 368 - 826 $\mathrm{kms^{-1}}$, respectively. The Alfvén Mach number is of the order of $1.2 \leq \mathrm{M_A} \leq 1.8$ at the above-mentioned heliocentric distance. In addition, the measured magnetic field strength is consistent with the earlier reports and follows a single power-law $B(r) = 6.07r^{-3.96}\ G$. Based on the current analysis, it is found that 19 out of 32 type II bursts are associated with immediate space weather events in terms of radio blackouts and polar cap absorption events, making them strong indications of space weather disruption. The ROTI enhancements, which indicate ionospheric irregularities, strongly correlate with GOES X-ray flares, which are associated with type II radio bursts recorded. The diurnal variability in ROTI is proportional to the strength of the associated flare class, and the corresponding longitudinal variation is attributed to the difference in longitude. This article demonstrates that since type II bursts are connected to space weather hazards, understanding various physical parameters of type II bursts helps to predict and forecast the space weather.

## 1 Introduction

The interaction of coronal mass ejections (CMEs) and their shocks with the magnetosphere is the major cause of the strongest space weather events in the magnetosphere. Shocks can be observed at extreme ultraviolet, visible and radio wavelengths

(Maguire et al., 2020; Carley et al., 2021). CMEs trigger space weather hazards by compressing the Earth's magnetosphere upon their arrival at the Earth which results in channeling the particles into the Earth atmosphere to produce Auroras. CMEs are also responsible for geomagnetic storms, power grid disruptions, accelerating solar energetic particles (SEPs) events etc. The energy released by explosive flares produces disruptions in the Earth's atmosphere within 8 minutes of the initial emission time (Salmane et al., 2018; Vourlidas et al., 2020). Solar flare radiations interact with ionosphere constituents, causing an immediate

rise in total electron density of the ionosphere. The extent of ionospheric total electron content (TEC) enhancements appears to be determined by the category of solar flares (Liu et al., 2004, 2006; Kumar and Singh, 2012; Al-Awadi et al., 2023). During the peak of X-ray solar flare, ionospheric TEC abnormalities are frequently suppressed due to accelerated solar energetic particles (Oljira, 2023). The rate at which TEC varies temporally is related to the effective flare radiation flux (Wan et al., 2002). Enhanced TEC in the ionospheric D-layer causes the absorption and blocking of high-frequency radio signals, resulting in

significant radio blackouts (R3) (Al-Awadi et al., 2023). Radio blackouts are disruptions in wireless communication and global positioning satellite (GPS) systems that use radio waves to communicate through the ionosphere. The National Oceanic and Atmospheric Administration (NOAA) classifies radio blackouts into five levels, which occur when radio signals carried through the ionosphere are reduced or absorbed (Kumar and Singh, 2012). In addition to Global navigation satellite system (GNSS) signals, ionospheric disturbances have a significant impact on high frequency communications. Pi et al. (1997) developed an

index known as the Rate Of change of TEC (ROT), that is based on the time rate of various phase changes in dual-frequency GNSS signals crossing the same ionospheric parcel and is expressed in TECU/min ($1\ TECU = 10^{16}\ electrons/m^2$). Depending on the dual-frequency GPS signals, ROT explains the irregularities on various length scales. The standard deviation of the ROT is used to construct the Rate Of TEC Index (ROTI) which has the same unit as ROT (e.g., Pi et al., 1997; Cherniak et al., 2014; Liu et al., 2019). ROTI describes the small -scale irregularity of the line of sight electron content resulting from

the ionosphere (Pi et al., 1997; Liu et al., 2019). During the solar minimum, the corotating interaction regions (CIRs) are the principal source of energetic particles in the heliosphere (e.g., McDonald et al., 1976; Van Hollebeke et al., 1978; Richardson et al., 1993). CIRs develop when a stream of rapid solar wind emerges from a coronal hole that reaches to low latitudes and overtakes a parcel of slow solar wind generated by the Sun at earlier times. The solar rotation causes these plasmas of different speeds to become radially aligned and interact (e.g., Gosling and Pizzo, 1999). This interaction generates a compression area

that revolves with the Sun and can amplify to produce shocks that accelerate particles.

The radio emissions that occur in the solar atmosphere to interplanetary space arise from a broad range of physical phenomena with space weather implications (e.g., flares, solar energetic particles, CMEs and shocks, Fleishman et al., 2020; Nindos, 2020; Vourlidas et al., 2020). Solar radio bursts (SRBs) originate from different altitudes in the solar atmosphere and they are

50 observed over a wide range of wavelengths from millimeters to decameters. Plasma density, electron beam density, injected electron beam speed, local turbulence levels, etc, all have a significant impact on the generation of various solar radio bursts (Sasikumar Raja et al., 2022a). Furthermore, electron density, magnetic field, and turbulence levels change with the solar cycle phase (Sasikumar Raja et al., 2019; Ndacyayisenga et al., 2021; Sasikumar Raja et al., 2021). Moreover, it is obvious that the phase of solar activity affects the multiple coronal features outlined above, which in turn influences the formation of SRBs.

Wild et al. (1963) classified SRBs into five types according to their morphologies of their dynamic spectra and their origin. Of the five types, type II, III and IV bursts are relevant to space weather study because they are associated with space weather drivers, such as shock waves (type II bursts, Cairns et al., 2003; Cane and Erickson, 2005; Chernov and Fomichev, 2021), streams of electrons propagating along open magnetic field lines (type III bursts, Reid and Ratcliffe, 2014, for a review) and CMEs or post-flare loops (type IV bursts, Nindos et al., 2008; Kumari et al., 2021). In the present paper, metric type II radio bursts observed from the ground by extended Compound Astronomical Low frequency Low cost Instrument for Spectroscopy and Transportable Observatory, herein, e-CALLISTO (Benz et al., 2005, 2009) are studied.

First discovered by Payne-Scott et al. (1947), type II radio bursts are among the most powerful events in the solar radio emission observed at metric wavelengths (e.g., Wild and McCready, 1950). At present, it is generally accepted that type II radio emissions are excited by magnetohydrodynamic (MHD) shock waves driven by solar flares, CMEs and fast plasma flow in the magnetic reconnection regions (Maia et al., 2000; Pick et al., 2006; Temmer et al., 2010; Grechnev et al., 2011; Vasanth et al., 2011; Kumari et al., 2017; Gopalswamy et al., 2018; Zucca et al., 2018; Chernov and Fomichev, 2021; Koval et al., 2023). Physical properties of metric type II radio bursts including but not limited to drift rate, starting frequency and duration are used to study the dynamics of the middle and upper solar corona. For example, the Alfvén Mach number, $M_A = V_S/V_A$, ($V_S$ and $V_A$, are shock and Alfvén speeds, respectively) is calculated using three different methods: (i) from shock geometry in EUV images, (ii) from the ratio of the CME speed to the Alfvén speed and (iii) using shock parameters derived from type II radio band-splitting phenomena (Vršnak et al., 2002; Maguire et al., 2020; Koval et al., 2023). A recent study by Maguire et al. (2020) showed that these three methods give consistent results after their comparative analysis.

By analyzing one or two events, many authors (e.g., Cho et al., 2013; Cunha-Silva et al., 2015; Kumari et al., 2017; Maguire et al., 2020; Lata Soni et al., 2021; Kouloumvakos et al., 2021; Mann et al., 2022) have determined the magnetic field strengths and examined the spatial and temporal evolution of shock properties, as well as the conditions responsible for type II radio emissions during high solar activity of solar cycle 24. There have been few works completed during the rise and fall phases of solar cycle 24 (e.g., Gopalswamy and Yashiro, 2011; Vasanth et al., 2014). Kim et al. (2012), on the other hand, covered the entire solar cycle 23. In the current study, a number of events are analyzed during the ascending phase of solar cycle 25 which started in December 2019 (Kallunki et al., 2021; Ahluwalia, 2022; Brajša et al., 2022). In this article, we apply the Rankine-Hugoniot density jump relation and parameters of type II radio bursts to estimate the parameter of shock waves (shock and the Alfvén speed, the Alfvén Mach number ) of metric type II radio bursts observed by e-CALLISTO and then analyze their space weather implication in terms of the ionospheric TEC enhancements using ROTI variability on daily basis.

## 2 Observation

### 2.1 Type II Radio bursts observation

The radio data presented in the current work were observed by e-CALLISTO from May 2021 to December 2022 of solar cycle 25. Firstly, we selected a number of radio events from the observations made by the instrument (https://e-callisto.org) and selected 32 well separated type II radio bursts whose morphologies are clear. We then examined their association with the

current solar phenomena to give insights on the status of the ascending phase of the solar cycle 25. In order to investigate the implications of space weather in terms of TEC, each selected type II radio burst was associated with a coronal mass ejection (CME) and an onset of a solar flare. The flare records were checked from the solar monitor (website: https://solarmonitor.org/).The CME parameters were taken from the Large Angle and Spectrometric COronagraph (LASCO C2) on board the Solar and Heliospheric Observatory (SOHO, Brueckner et al., 1995) catalog updated to 30 December 2022.

## 2.2 Derivation of shock characteristic parameters

In this study, first we measured the bandwidth (BDW) of each type II radio burst. Because all of the type II bursts do not show band-splitting feature, the BDW of the fundamental band is linked to the ambient density jump to ensure consistency in computation.

$$BDW = \frac{f_u - f_l}{f_l} \tag{1}$$

where $f_u$ and $f_l$ denote the upper and lower frequencies, respectively of the fundamental emission band. Figure 1 shows an example of type II radio burst from 03:28:25 UT to 03:32:30 UT on April 17, 2022 for which $f_u$ and $f_l$ are indicated. This burst is associated with an X1.1 flare that started at 03:17 UT and stopped at 03:51 UT from NOAA active region 12994. The values of BDW were used to estimate the density jump across the shock (Vršnak et al., 2001, 2002; Cho et al., 2007; Nedal et al., 2019), $\chi$ via the relation

$$\chi = (BDW + 1)^2 \tag{2}$$

By assuming low plasma ratio ($\beta \to 0$) for a perpendicular shock in the corona (Vršnak et al., 2001, 2002), the density jump allows one to compute the Alfvén Mach number ($M_A$) using Rankine-Hugoniot approximation

$$M_A = \sqrt{\frac{\chi(\chi + 5)}{2(4 - \chi)}} \tag{3}$$

It has been shown that the rate of change of the frequency of metric type II radio bursts is related to the shock speed and the electron density gradient in the solar corona (e.g., Gopalswamy, 2011; Vemareddy et al., 2022) via the following

$$V_s = -\frac{2r}{\alpha}\left(\frac{1}{f}\right)\left(\frac{df}{dt}\right) \tag{4}$$

where $r$ is the shock formation height, $\alpha$ is a fitted empirical index of density variation over the heliocentric distance, $f$ is the starting frequency, and $\frac{df}{dt}$ is the frequency drift rate. The electron density decreases with heliocentric distance from the Sun, according to the power-law: $n_e(r) \propto r^{-\alpha}$. Three different density models by Newkirk (1967), Saito et al. (1977) and Leblanc et al. (1998) describe the variation of the electron density in the corona and interplanetary medium. With these models, it has been observed that within $1 - 3$ solar radii ($R_\odot$), the electron density is directly proportional to $r^{-6}$ in the corona and directly proportional to $r^{-2}$ beyond few tens of solar radii. Because the type II radio observed all have occurred in the range of $\sim 1 - 2\,R_\odot$, $\alpha$ is chosen to be 6.13 (Gopalswamy, 2011). The Alfvén velocity is directly related to the shock speed as

$$V_A = \frac{V_s}{M_A} \tag{5}$$

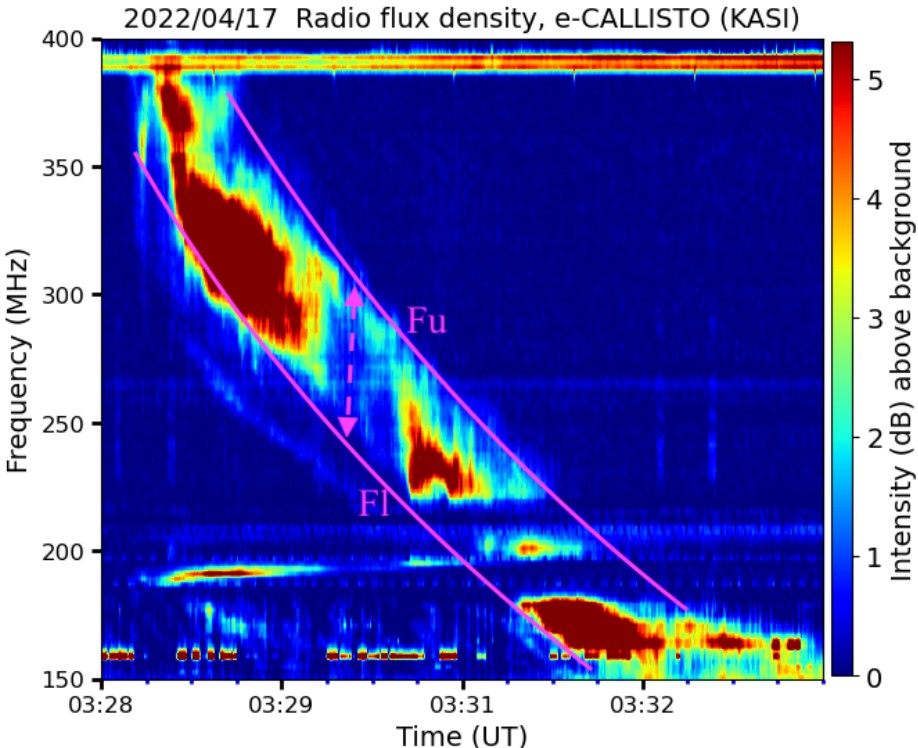

**Figure 1.** Type II radio burst from 03:28:25 UT to 03:32:30 UT observed by Korean Astronomy and Space science Institute (KASI) on April 17, 2022. $F_u$ and $F_l$ denote the upper and lower frequencies of the fundamental band of the type II radio emission.

In the region surrounding a CME, the ambient magnetic field strength ($B$) of the plasma can be estimated using the relation (Vršnak et al., 2002; Cho et al., 2007; Lata Soni et al., 2021)

$$B[G] = 5.1 \times 10^{-5} \times f_l[MHz] \times V_A[km/s] \tag{6}$$

where $f_l$ is the lower frequency of the fundamental frequency band.

### 2.3 Ionospheric data and Solar Energetic Particles

Data from ground-based GPS receiver stations around the world were used to analyze the ionospheric TEC for disturbed days identified by type II radio burst observations in this study. These include the African Geodetic Reference Frame (AFREF) database (http://afrefdata.org/) and UNAVCO Archive of GNSS Data (https://www.unavco.org/). Figure 2 maps the geographic locations of some GNSS stations used in the current study for a reference. As GPS data are usually provided in Receiver Independent EXchange (RINEX) format, TEC were derived from RINEX files using the GPS TEC software developed at Boston college, assuming a thin shell ionosphere at the altitude of 350 km. Details on the software used to derive TEC are provided in Seemala and Valladares (2011); Uwamahoro et al. (2018, and references therein). To reduce the multipath effects

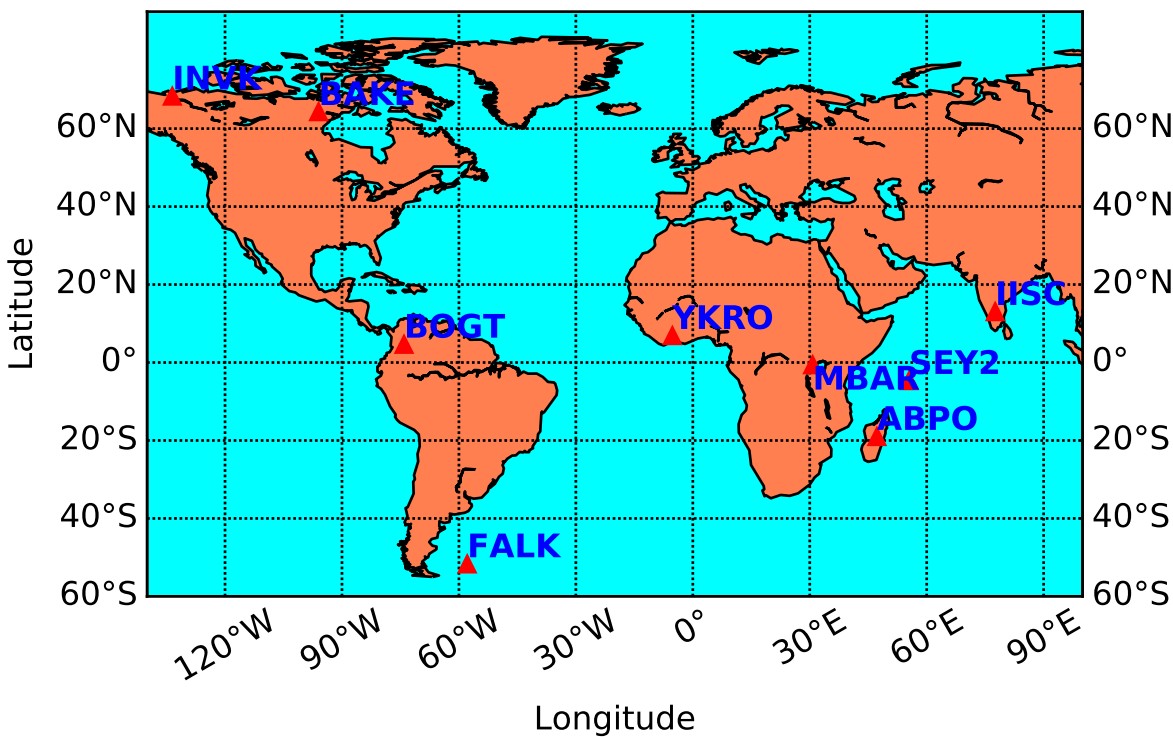

**Figure 2.** Geographic locations of some GNSS stations Codes used for this study (FALK: Falkland, Islands; ABPO: Madagascar; MBAR: Uganda; SEY2: Seychelles; IISC: India; YKRO: Ivory coast; BOGT: Colombia; BAKE: Canada and INVK: Canada).

on slant TEC (STEC), the elevation angle was fixed to $30°$. The ROT was calculated using the equation 7 proposed by Pi et al. (1997), and has been utilized by several researchers to explore ionospheric irregularities (Azzouzi et al., 2015; Liu et al., 2016, 2019; Dugassa et al., 2020; Habyarimana et al., 2023).

$$ROT = \frac{STEC_{k+1} - STEC_k}{\Delta t_k} \tag{7}$$

where $STEC_{k+1}$ and $STEC_k$ are the STEC values at two successive epochs, and $\Delta t_k$ is the time difference between them

equivalent to 30 s for IGS given in Figure 2. Equation 8 was used to calculate the ROTI, which was defined as the standard deviation of ROT over 5 minutes.

$$ROTI = \sqrt{< ROT^2 > - < ROT >^2} \tag{8}$$

where the $<>$ stands for the time average value. The solar energetic particles were taken from the database at website (https://cdaweb.gsfc.nasa.gov/, last accessed on 19 February 2024).

## 3 Results and Discussions

### 3.1 Comparison and analysis

During the ascending phase of the solar cycle 25, the e-CALLISTO network observed a series of solar radio bursts in the range from 5 to 870 MHz. With the interest of space weather diagnostics, 32 well separated type II radio bursts observed are presented in this study. Table 1 lists the spectrometers used in this study, as well as their geographic locations, frequency range of observation, and number of radio bursts taken at each station. All spectrometers are observing in a narrow frequency range of few tens of MHz. Using the radio parameters such as bandwidth, drift rate and starting frequency, the shock characteristics

**Table 1.** e-CALLISTO Spectrometers, their geographical locations and their frequency ranges.

| # | File ID | Country | Lat(deg) | Long (deg) | Obs. Frequency range (MHz) | # of events |
|---|---------|---------|----------|------------|----------------------------|-------------|
| 1 | Australia-ASSA | South Australia | -34.66 | 139.64 | 15 — 87 | 11 |
| 2 | Arecibo-Observatory | Puerto Rico, USA | 18.22 | -66.59 | 15 — 87 | 9 |
| 3 | GREENLAND | Greenland | 67.00 | -50.72 | 10 — 110 | 3 |
| 4 | ALASKA-HAARP | ALASKA | 62.40 | -150.20 | 5 — 87 | 2 |
| 5 | ALMATY | Kazakhstan | 43.22 | 76.83 | 45 — 165 | 1 |
| 6 | BIR | Ireland | 16.61 | 77.51 | 10 — 100 | 2 |
| 7 | INDIAN-OOTY | India | 11.41 | 76.69 | 45 — 165 | 1 |
| 8 | KASI | South Korea | 36.35 | 127.38 | 150 — 400 | 1 |
| 9 | MEXICO-LANCE | MEXICO | 19.81 | -101.69 | 50 — 90 | 1 |
| 10 | SWISS-Landschlacht | Switzerland | 47.63 | 9.25 | 15 — 87 | 1 |

from each radio event have been estimated. Table 2 illustrates each type II radio burst selected and associated CME , GOES soft X-ray flares as well as shock characteristics. The first column of this table is the numbering index of the events, the next four columns are the date of the radio events in the format dd/mm/yyyy hh:mm, their starting frequencies, f (MHz), their drift rates (MHz/s) and their shock formation heights ($R_\odot$) estimated using the relation $f(r) = 307.87r^{-3.78} - 0.14$ (Gopalswamy et al., 2013). Columns 6 to 9 are the GOES soft X-ray flares parameters (start, class, NOAA region and location) followed by two columns that present the CME onset and speed, respectively. Columns 12 to 15 present the shock characteristics (density jumps, Mach numbers, shock and Alfvén velocities, respectively) while the last column of this table presents the estimated ambient magnetic field strength, B (Gauss). There is a strong correlation (CC=0.98) between the drift rates and starting frequencies of the type II radio bursts (Figure 3) which are the key parameters to estimate the shock speeds from the dynamic spectra. Higher starting frequency have higher drift rates (Umuhire et al., 2021). Such correlation agrees well with the previous studies, giving slopes of $\epsilon = 1.89$ and $\epsilon = 1.33$, respectively (e.g., Vršnak et al., 2002; Umuhire et al., 2021). From Table 2, it is clearly observed that 4 out of 32 radio events are not associated with any solar flare because they originate from the farside on the solar surface but the shocks generating these bursts were excited by associated CMEs. It is also noticed that 19/28 are connected with intense GOES X-ray flares (M & X classes), which is compatible with their speeds as well as estimated shock

**Table 2.** Type II radio bursts observed by e-CALLISTO during the ascending phase of solar cycle 25 and their associated CMEs, GOES soft X-ray flares and estimated shock characteristics.

| No | Type II burst event | | | | Soft X-ray flare | | | | CME | | Shock characteristics | | | | B-field |
|---|---|---|---|---|---|---|---|---|---|---|---|---|---|---|---|
| | Date | f | Drift rate | height | Start | Class | NOAA | Location | Onset | Speed | $\chi$ | $M$ | $V_s$ | $V_A$ | |
| | (UT) | (MHz) | (MHzs$^{-1}$) | R$_\odot$ | (UT) | | | | (UT) | (kms$^{-1}$) | | | (kms$^{-1}$) | (kms$^{-1}$) | G |
| 1 | 22/05/2021 02:57 | 86 | - 0.13 | 1.4 | 02:47 | C6.1 | 12824 | N18E25 | ...... | ..... | 1.6 | 1.5 | 752 | 504 | 1.5 |
| 2 | 23/06/2021 07:05 | 73 | -0.10 | 1.5 | 06:43 | C3.4 | 12833 | N14E89 | 07:24 | 390 | 1.5 | 1.4 | 668 | 464 | 1.2 |
| 3 | 25/07/2021 04:54 | 64 | -0.11 | 1.5 | ..... | .... | F. S. | ...... | 05:48 | 237 | 1.3 | 1.2 | 785 | 637 | 1.6 |
| 4 | 28/08/2021 05:10 | 64 | -0.11 | 1.5 | 05:01 | C7.0 | 12860 | S31E06 | ..... | ..... | 1.7 | 1.6 | 894 | 556 | 1.2 |
| 5 | 09/10/2021 06:34 | 75 | - 0.11 | 1.5 | 06:19 | M1.6 | 12882 | N18E06 | 07:00 | 712 | 1.6 | 1.5 | 735 | 496 | 1.3 |
| 6 | 09/10/2021 06:49 | 31 | - 0.04 | 1.9 | 06:19 | M1.6 | 12882 | N18E06 | 07:00 | 712 | 1.3 | 1.3 | 706 | 561 | 0.7 |
| 7 | 28/10/2021 15:28 | 90 | -0.18 | 1.4 | 15:17 | X1.0 | 12887 | S26W07 | 15:48 | 1519 | 2.0 | 1.8 | 1273 | 697 | 1.6 |
| 8 | 20/12/2021 11:27 | 87 | - 0.14 | 1.4 | 11:12 | M1.8 | 12908 | S20W01 | 12:36 | 386 | 1.7 | 1.6 | 750 | 479 | 1.5 |
| 9 | 12/01/2022 04:28 | 69 | - 0.11 | 1.5 | ..... | .... | F. S. | ...... | 03:12 | 433 | 1.8 | 1.7 | 816 | 479 | 1.1 |
| 10 | 12/02/2022 08:33 | 173 | - 0.36 | 1.2 | 08:25 | M1.4 | 12939 | S17W82 | 08:12 | 785 | 1.3 | 1.2 | 792 | 659 | 4.1 |
| 11 | 02/03/2022 17:42 | 67 | - 0.11 | 1.5 | 17:31 | M2.0 | 12958 | N15E29 | 18:24 | 248 | 1.9 | 1.7 | 924 | 532 | 1.1 |
| 12 | 14/03/2022 17:20 | 98 | - 0.13 | 1.4 | 17:13 | B8.5 | 12964 | S30W86 | 17:48 | 534 | 1.9 | 1.7 | 883 | 506 | 1.2 |
| 13 | 25/03/2022 05:15 | 66 | - 0.12 | 1.5 | 05:02 | M1.4 | 12974 | S18E37 | 06:12 | 433 | 1.5 | 1.4 | 801 | 590 | 1.6 |
| 14 | 28/03/2022 11:23 | 87 | - 0.15 | 1.4 | 10:58 | M4.0 | 12975 | N18W04 | 12:12 | 335 | 1.8 | 1.7 | 951 | 554 | 1.4 |
| 15 | 30/03/2022 17:33 | 72 | - 0.11 | 1.5 | 17:21 | X1.3 | 12975 | N13W31 | 18:00 | 493 | 1.9 | 1.8 | 1128 | 654 | 1.1 |
| 16 | 31/03/2022 18:34 | 67 | - 0.13 | 1.5 | 18:17 | M9.6 | 12975 | N12W47 | 19:12 | 489 | 2.0 | 1.8 | 1081 | 594 | 1.3 |
| 17 | 02/04/2022 13:24 | 71 | - 0.15 | 1.5 | 12:56. | M3.9 | 12975 | N12W68 | 13:36 | 686 | 1.8 | 1.6 | 1038 | 631 | 1.5 |
| 18 | 17/04/2022 03:28 | 382 | - 0.83 | 0.9 | 03:17 | X1.1 | 12994 | N12E88 | 03:48 | 728 | 1.2 | 1.2 | 828 | 711 | 7.8 |
| 19 | 21/04/2022 02:00 | 85 | -0.15 | 1.4 | 01:47 | M9.6 | 12993 | N22E23 | 02:36 | 828 | 1.7 | 1.5 | 1070 | 696 | 1.6 |
| 20 | 21/04/2022 22:47 | 69 | -0.11 | 1.5 | 22:39 | C1.6 | 12993 | N12E25 | 23:12 | 389 | 1.4 | 1.3 | 791 | 591 | 1.4 |
| 21 | 30/04/2022 13:46 | 83 | -0.13 | 1.4 | 13:37 | X1.1 | 12994 | N16W88 | 14:00 | 535 | 1.7 | 1.5 | 936 | 610 | 1.4 |
| 22 | 30/04/2022 19:50 | 80 | -0.12 | 1.4 | 19:42 | M1.9 | 12994 | N16W88 | 20:12 | 793 | 1.7 | 1.6 | 855 | 543 | 1.3 |
| 23 | 04/07/2022 13:35 | 69 | -0.13 | 1.5 | 12:23 | C5.1 | 13050 | N17E36 | 11:36 | 256 | 1.7 | 1.6 | 918 | 581 | 1.4 |
| 24 | 05/07/2022 04:16 | 69 | -0.10 | 1.5 | 03:59 | C9.8 | 13045 | S20W18 | 05:00 | 515 | 1.6 | 1.5 | 761 | 512 | 1.2 |
| 25 | 14/08/2022 12:05 | 70 | -0.08 | 1.5 | 11:50 | C2.4 | 13076 | N21W14 | 13:25 | 411 | 1.4 | 1.3 | 512 | 402 | 1.1 |
| 26 | 18/08/2022 12:12 | 62 | -0.16 | 1.6 | ..... | .... | F. S. | ...... | 11:00 | 1131 | 1.7 | 1.6 | 1282 | 826 | 1.9 |
| 27 | 19/08/2022 04:35 | 81 | -0.10 | 1.4 | 04:14 | M1.6 | 13078 | S27W48 | 04:49 | 695 | 1.3 | 1.2 | 504 | 420 | 1.4 |
| 28 | 23/09/2022 18:02 | 67 | -0.12 | 1.5 | 17:48 | M1.7 | 13110 | N16E84 | 18:12 | 687 | 2.0 | 1.8 | 1010 | 548 | 1.1 |
| 29 | 29/09/2022 12:06 | 80 | -0.10 | 1.4 | 11:50 | C5.7 | ..... | N26E86 | 12:24 | 321 | 1.5 | 1.4 | 672 | 473 | 1.2 |
| 30 | 09/11/2022 20:03 | 89 | -0.11 | 1.4 | ..... | .... | F. S. | ...... | 20:36 | 371 | 1.5 | 1.4 | 618 | 435 | 1.3 |
| 31 | 03/12/2022 17:44 | 84 | -0.13 | 1.4 | 17:36 | M1.2 | 13157 | N14E89 | ..... | ..... | 1.8 | 1.8 | 857 | 518 | 1.3 |
| 32 | 14/12/2022 08:30 | 160 | -0.22 | 1.2 | 08:24 | M1.1 | 13162 | S16W89 | 08:48 | 402 | 1.9 | 1.8 | 657 | 368 | 1.7 |

speeds. We derived the shock and Alfvén speeds of these type II radio bursts in the order of 504 - 1282 kms$^{-1}$ and 368 - 826 kms$^{-1}$, respectively at heliocentric distance $\sim 1-2$ $R_\odot$. Comparatively, values are consistent with the measurements reported by Cunha-Silva et al. (2015); Minta et al. (2023) about 590 - 810 kms$^{-1}$ and 250 - 550 kms$^{-1}$, respectively at $\sim 1.2-1.8$ $R_\odot$. The Alfvén speeds from the current work are also in agreement with the range of the Alfvén speeds of 140 – 460 kms$^{-1}$ over 1.2 – 1.5 $R_\odot$ and 259 – 982 kms$^{-1}$ over 3 - 15 $R_\odot$ given in Gopalswamy and Yashiro (2011) and in Kim et al. (2012), respectively. Figure 4 presents the correlation between the speeds from the LASCO field of view (FOV) and the speeds derived from the dynamic spectra. Table 2 observations and Figure 4 show that there are estimated shock speeds that are faster than CME speeds from LASCO FOV and vice versa. The difference in CME speed between dynamic spectra and LASCO is at-

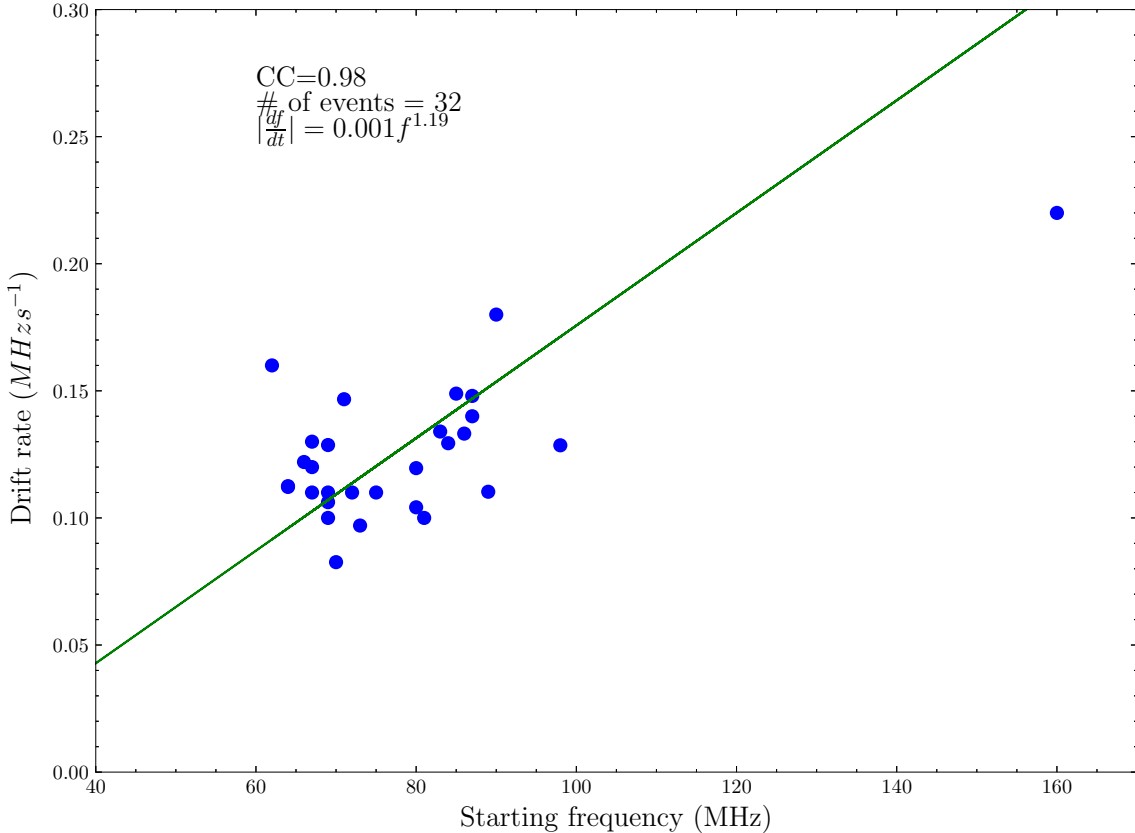

**Figure 3.** Scatter plot between the absolute drift rates ($|\frac{df}{dt}|$) and the starting frequency ($f_s$) for all the 32 metric type II radio bursts. The power-law least squares fits and the corresponding correlation coefficient (CC) are shown.

tributed to the CME's central position angle as observed by LASCO, implying that the shock may be weakened and dissipated before entering LASCO FOV (Gopalswamy et al., 2011). On the other hand, the shock decelerates in the case of a decline in its intensity or when it breaks. The type II burst only serves as a time marker for when the shock occurs. It should be noted that type II radio emission can come from anywhere on the shock front: the nose or the flanks, depending on which location is best for electron acceleration (Gopalswamy et al., 2013). Solar radio type II bursts associated with slow CMEs are thought to be generated from non-thermal electrons accelerated by a moving magnetic reconnection when slow CMEs interact with the background magnetized coronal plasma (Tan et al., 2019). Furthermore, a recent study confirmed that observing a type II radio burst is evidence of shock acceleration in the solar corona (Chernov and Fomichev, 2021).

The Alfvén Mach numbers in the range $\sim 1.2 - 1.8$ at $\sim 1 - 2\,\mathrm{R}_\odot$ are consistent with the measurements of about 1.1 - 1.9 at $\sim 1.3 - 2.5\,\mathrm{R}_\odot$ reported by Vršnak et al. (2002) and that of Cunha-Silva et al. (2015) in the order of 1.4 to 1.7 at $\sim 1.2 - 1.8\,\mathrm{R}_\odot$. The magnetic field strength is an important parameter that influences the dynamical eruption of CMEs in the solar atmosphere

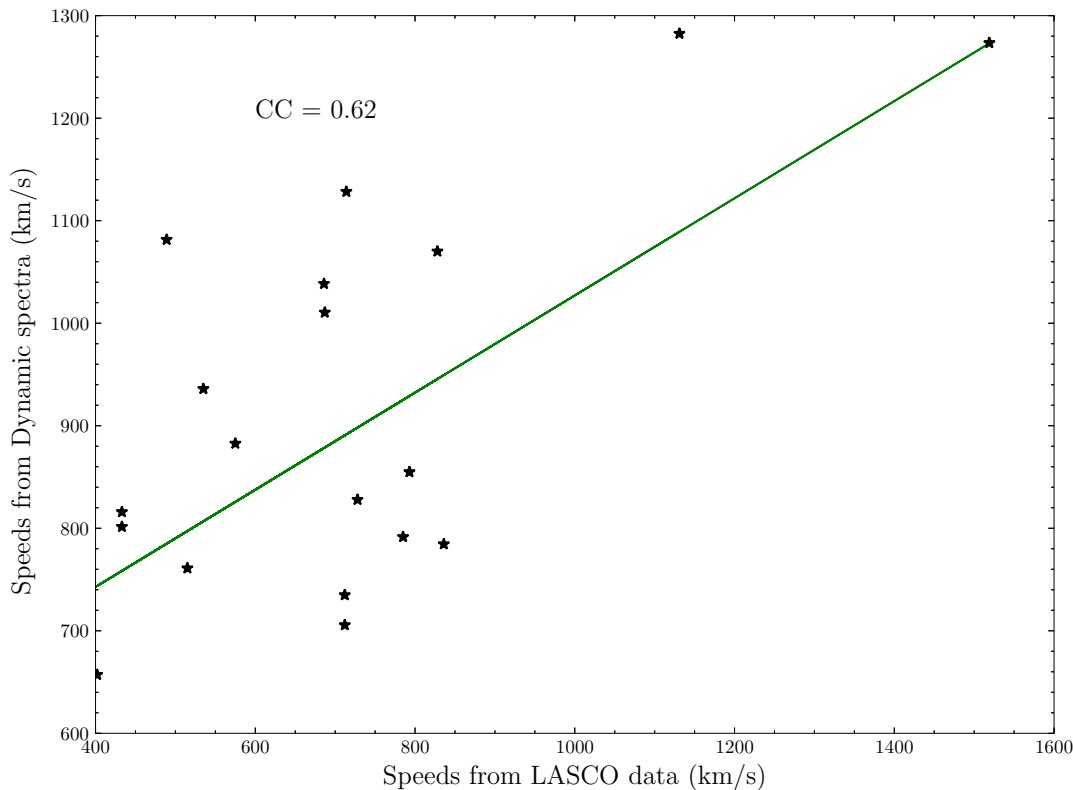

**Figure 4.** Scatter plot showing the correlation between the speeds from LASCO FOV and speeds derived from dynamic spectra. Higher values of speeds obtained from dynamic spectra are attributed to the radio source, which propagates at faster speeds due to the interaction of slow CMEs with background magnetized coronal plasma (Tan et al., 2019).

(Sasikumar Raja et al., 2014; Carley et al., 2017). High-starting type II radio bursts are associated with coronal shocks that are closer to the solar surface. As a result, high magnetic field values are expected. Figure 5 demonstrates the variation of the magnetic field strength estimated in this study (Equation 6) relative to the quiet Sun magnetic field model $B(r) = \frac{a}{r^2}$ with $a = 2.2$ (Gopalswamy et al., 2001) and Dulk and McLean (1978) empirical model for the magnetic field above active region 185 $B(r) = 0.5 (r-1)^{-1.5}$. The magnetic field has been calculated in the range $0.5 < B < 8\ G$ at $\sim 1 - 2\ \mathrm{R_\odot}$, which shows excellent consistency with earlier researches and is fitted with a single power-law distribution of the type $B(r) = 6.07 r^{-3.96}\ G$ as represented by the black dotted curved of Figure 5. However, Rankine - Hugoniot jump relation has been used by a number of researchers to derive shock parameters. For example, with this technique Smerd et al. (1974, 1975) found $1.2 \leq \mathrm{M_A} \leq 1.7$ and $0.3 \leq \mathrm{B} \leq 4\ G$. The same technique was applied by Vršnak et al. (2002) who reported a magnetic field strength in the 190 range 1 - 8 G at heliocentric distance of $\sim 1.6\ \mathrm{R_\odot}$. A field strength of 6 - 5 G at $\sim 1.5 - 1.77\ \mathrm{R_\odot}$ is reported by Ramesh et al. (2010). Dulk and McLean (1978); Sasikumar Raja et al. (2022b) have given a detailed review on solar coronal magnetic fields measured using different techniques and at different wavelengths of the electromagnetic spectrum. A recent work has

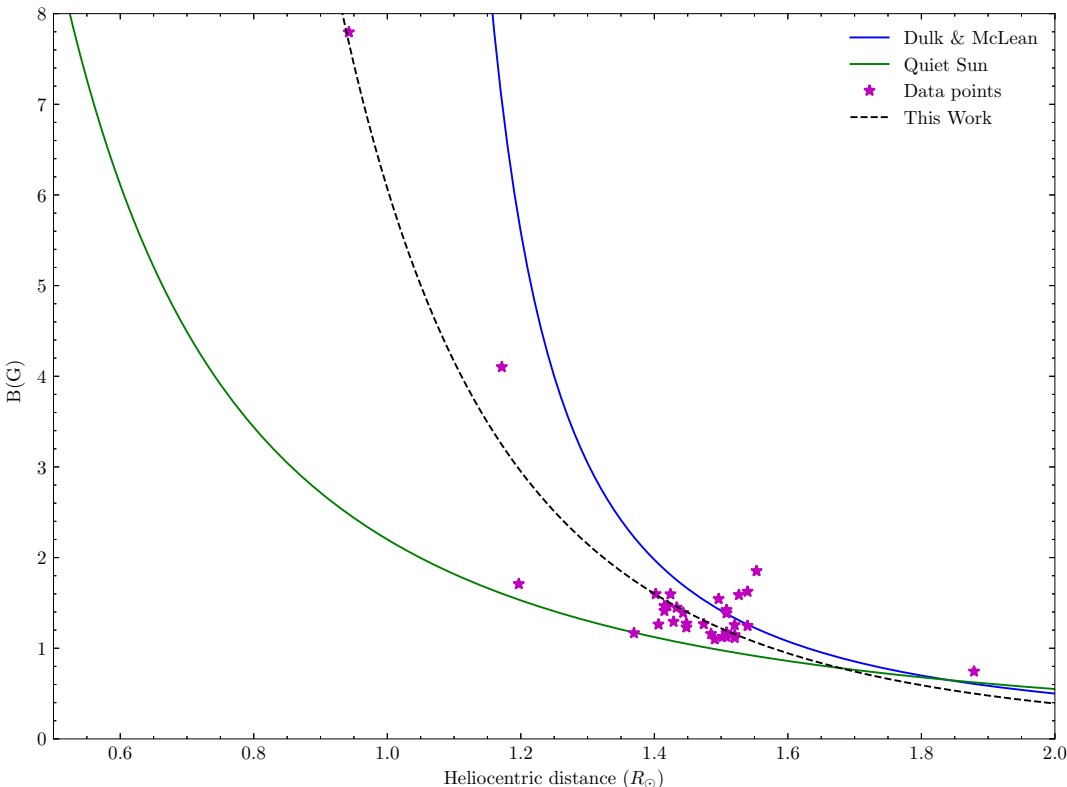

**Figure 5.** Comparison of the magnetic field strength from the current study, the quiet Sun magnetic field model (Gopalswamy et al., 2001) and the empirical magnetic field relation (Dulk and McLean, 1978). The magnetic values estimated are all above the quiet Sun magnetic model and the pattern is close to the empirical model which confirms that the Sun was awake.

reported that two necessary conditions for type II radio emissions: (i) relatively intense shock waves (the Mach number should exceed a certain value $M_{cr}$) and (ii) perpendicular shock waves are required (Chernov and Fomichev, 2021). Our values of Mach numbers $1.2 \leq M_A \leq 1.8$ agrees well with these conditions. In Table 3, the statistical findings from this study and earlier research that examined more than two radio events are summarized and compared.

**Table 3.** Comparison of the statistical findings of this study and previous studies that analyzed more than two radio events.

| Epoch | # of events | Mean shock speed ($km/s$) | Mean Alfvén speed | B-field range (G) | Height range ($R_\odot$) | Citation |
|---|---|---|---|---|---|---|
| 2021 – 2022 | 32 | 860 | 566 | 8 – 0.5 | 1.0 — 2.0 | This work |
| 2013 – 2014 | 4 | 739 | 579 | 1.8 – 1.3 | 1.7 – 1.9 | Kishore et al. (2016) |
| 1996 – 2007 | 10 | 1288 | 555 | 0.105 – 0.006 | 3 – 15. | Kim et al. (2012) |

## 3.2 Associated Space weather implication

The ascending phase of solar cycle 25 is characterized by more intense solar activity than expected (e.g., Dang et al., 2022; Hapgood et al., 2022; Kataoka et al., 2022). Tan (2011) and Sarp et al. (2018) show that solar cycle 25 is more active than the previous cycle and is more consistent with actual observations as predicted. Furthermore, Du (2020) estimated that the maximum peak of cycle 25 would be 30% stronger than that of cycle 24. These indicate that the activity would be high, and we use this advantage to track the intensity of early space weather events in the current cycle. To account for ionospheric irregularities caused by concurrent GOES X-ray flares, Type II solar radio bursts were utilized as selection criteria of disturbed days due to their association with solar phenomena such as radio blackouts. The ROTI were examined on 25 type II radio bursts, which are linked to both solar flares and CMEs, by selecting GNSS stations in either equatorial, mid-latitude, and high-latitude regions. Furthermore, the ROTI classifies the irregularities of the ionospheric TEC as no TEC irregularity (ROTI < 0.25 TECU/min), weak (0.25 ≤ROTI < 0.5 TECU/min), moderate (0.5 ≤ROTI < 1.0 TECU/min) and strong (ROTI ≥ 1.0 TECU/min) (Liu et al., 2016). It is worth noting that four major solar energetic particles (> 10 MeV, SEPs) occurred on days when type II radio bursts are observed, and these dates are used as illustrative examples in this study.

### 3.2.1 October 28, 2021 event

Type II radio bursts on 2021 October 28, was recorded by the CALLISTO spectrometer, Birr castle, Ireland. This type II burst is recorded in the time range of 15:28 UT to 15:38 UT which overlapped by a type IV radio burst from 15:32 UT to 15:43 UT as indicated in Figure 6. This radio event is associated with the GOES soft X-ray flare of X1.0 class that started at 15:17 UT, peaked at 15:35 UT and stopped at 15:38 UT from NOAA Active region (AR) 12887 explosion. It is also associated with energetic halo CME observed by LASCO C2 coronagraph with onset at 15:48 UT with a speed of 1519 kms$^{-1}$. This CME did not reach near Earth and therefore no geomagnetic storm was recorded in next five days. It is observed that few minutes after the type II had started, an enhancement of protons took place as an effect of radio blackout (R3: major, https://spaceweather.com/images2021/28oct21/blackout_x1.jpg) which affected the whole south America and Atlantic sea. Figure 7 depicts the ionospheric irregularities in terms of ROTI observed in different region of the globe. It is noted that the flare has no direct interaction with the magnetosphere but its radiation agents (X-rays, UV, EUV) perturb the ionosphere by increasing the ionization which in turn causes the signal delay in Global Navigation Satellite Systems (GNSS) (e.g., Amory-Mazaudier et al., 2017). Figure 7 clearly shows that there is no ionospheric perturbation associated with the X1.0 flare in the equatorial region over Seychelles (Figure 7 a), whereas ROTI is strongly suppressed in the mid-latitude zone over India (Figure 7 b). The ROTI profile in high latitude region over Colombia (Figure 7 C) is consistent with the X1.0 flare flux profile (Figure 7 d). According to Habarulema et al. (2022), the F2 layer was unaffected by the X1.0 flare on 28 October 2021. However, ROTI is strongly suppressed in equatorial region (SEY2) when substantial SEP arrives at 17:00 UT. On this day, a major SEP (> 10 MeV) is observed with an onset time of roughly 17:00 UT on the high energy detector (HED) on board Solar & Heliospheric Observatory (SOHO), Energetic and Relativistic Nuclei, and Electron Experiment (ERNE), as shown in Figure 8. However,

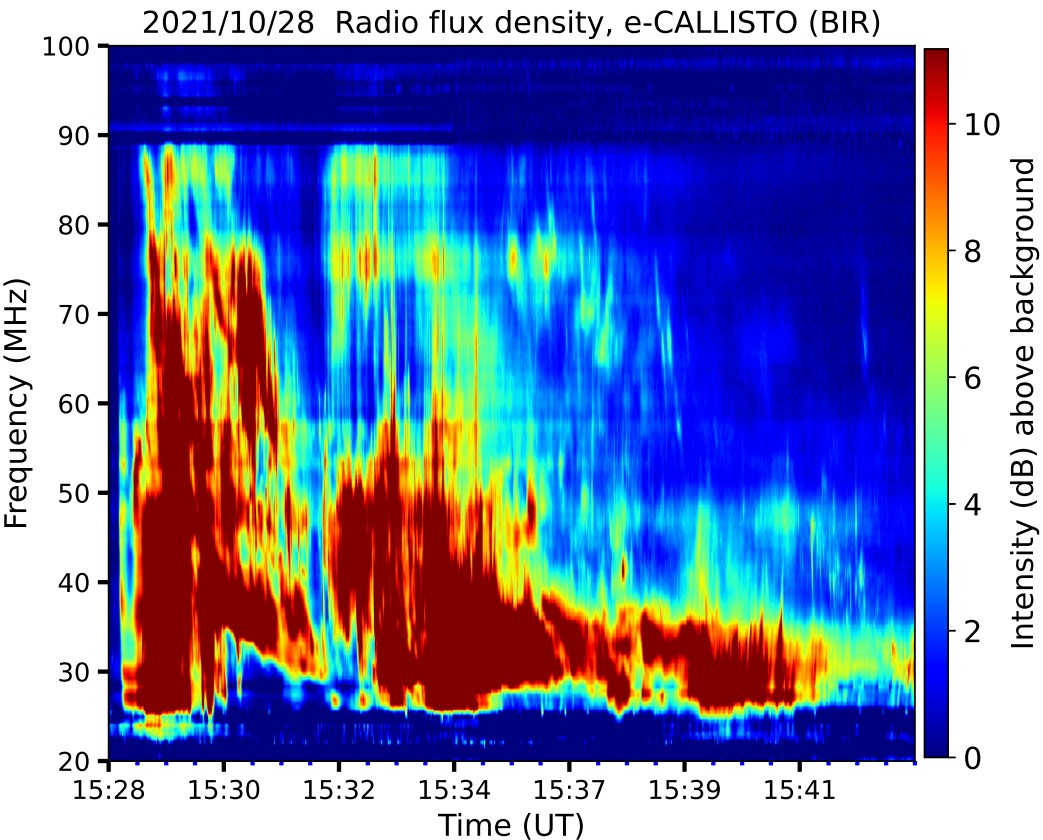

**Figure 6.** Type II radio emissions observed on 2021 October 28 overlapped by a type IV radio emission from 15:32 UT to 15:43 UT.

neutron monitor (https://gle.oulu.fi/) recorded the SEP on 28 October 2021 as ground level event (GLE) that started at 15:46
UT (Klein et al., 2022).

### 3.2.2    March 28, 2022 event

The solar activity is seen to be high during March 2022. This is due to a number of solar events observed and recorded during this month where seven type II radio events were recorded in March 2022. Figure 9 presents a type II radio burst observed by e-CALLISTO network at Arecibo Observatory in Puerto Rico, USA from 11:23:12 to 11:28:37 UT on 28 March 2022 with 87
235    - 32 MHz frequency range. This burst is overlapped by a type IV radio burst that occurred from 11:26 to 11:36 UT. These bursts are associated with GOES soft X-ray flare M4.0 that started at 10:58 UT, peaked at 11:29 UT and stopped at 11:45 UT from NOAA 12975. This eruption also produced a tsunami in the solar atmosphere (see, https://sdo.gsfc.nasa.gov/data/dailymov/movie.php?q=20220328_1024_0193). The bursts are also associated with a partial halo CME with speed of 335 $kms^{-1}$ and the CME was off Sun - Earth line because no geomagnetic storm is linked to it. However, the flare and the tsunami accelerated

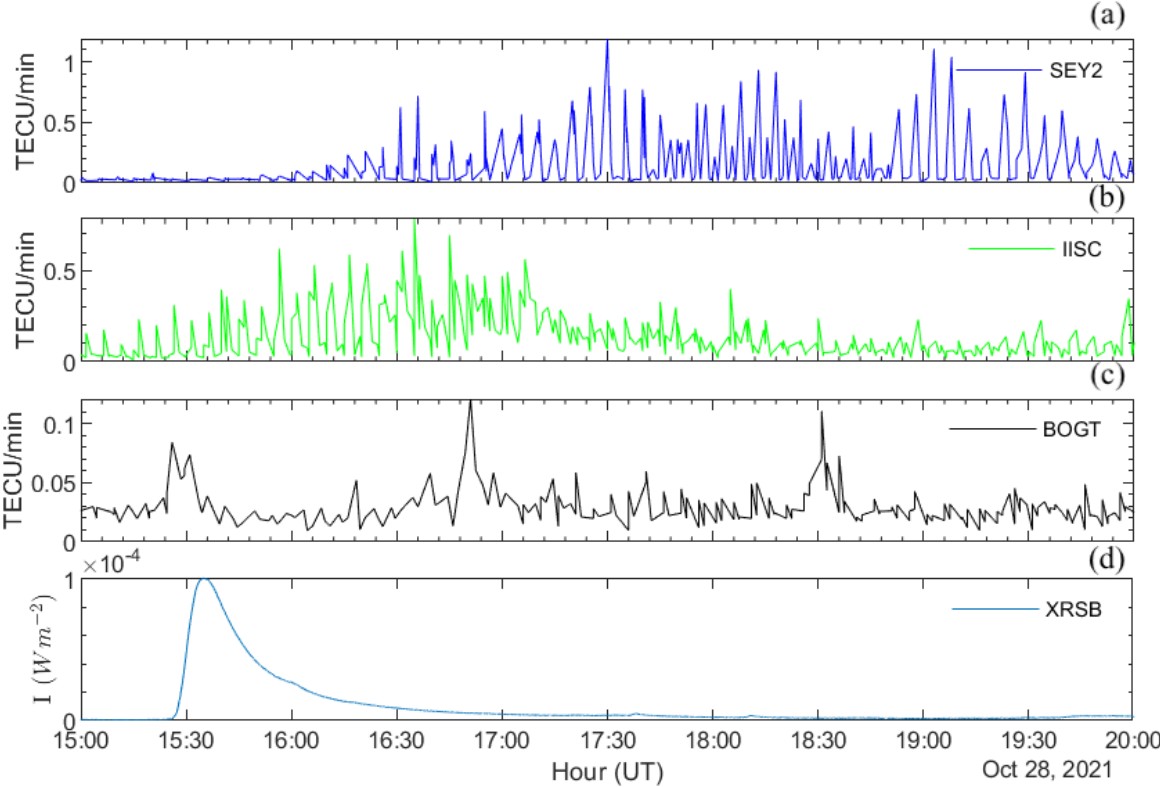

**Figure 7.** Variability of ROTI in (a) equatorial region (SEY2) (b) mid-latitude region (IISC) (c) high latitude region (BOGT) and (d) associated GOES soft X-ray X1.0 flare intensity profile.

protons that hit the Earth's magnetosphere and caused a minor radiation storm. The enhancement of proton events is revealed by the radio blackout that cover the whole African continent (https://spaceweather.com/images2022/28mar22/blackout.jpg) and the polar cap absorption event (PCAE) that occurred after about 2:40 UT from the burst onset (https://spaceweather.com/images2022/28mar22/pca.jpg). This event is a signature of solar proton enhancement where High Frequency (HF) and Very High Frequency (VHF) are absorbed while low and very low frequencies are reflected at low altitude. Previous works showed that solar flares that cause solar energetic particles (SEPs) are usually accompanied by radio bursts and noise storms that disturb the ionospheric TEC (Ranta et al., 1993); and mostly observed 20 minutes to 20 hours after the solar flare (Mitra, 1974; Kavanagh et al., 2004; Perrone et al., 2004). They also showed that SEPs and PCAEs are frequent close to the maximum solar cycle (Shea and Smart, 2002), but the solar cycle 25 is far from its maximum. Thus, these observations are the evidence of high solar activity during the ascending phase of the current sunspot cycle. It is important to note that the association of type II radio bursts with space weather drivers such as solar flares, SEPs and coronal mass ejections make them special for space weather

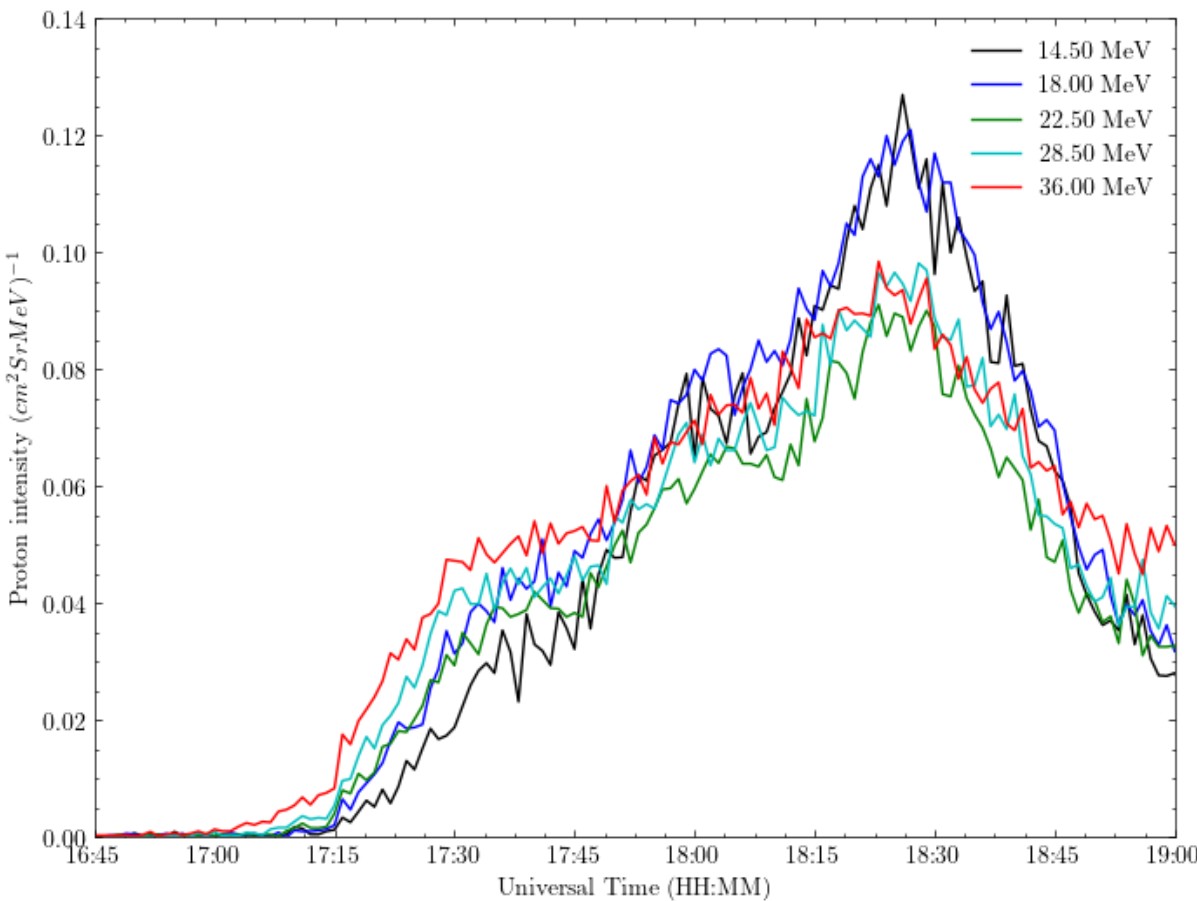

**Figure 8.** Profile of the particle intensity of the SEP on 28 October 2021 in five energy levels onset at 17:00 UT and peak at 18:26 UT.

(Kumari et al., 2019; Ndacyayisenga et al., 2021). Figure 10 presents the ionospheric irregularities using ROTI in response to the solar flare of 28 March 2022. Figure 10 (a) reveals strong TEC abnormalities (ROTI > 0.5 TECU/min) at the M4.0 flare intensity peak. In response to the M4.0 flare, no TEC anomalies are seen in mid-latitude and equatorial regions (Figure 10 (b - c)). Furthermore, the suppression of ROTI in the mid-latitude zone is related to the significant SEP, which began about 13:00 UT, as shown in Figure 11. This figure shows two peaks in particle intensities at 15:50 UT and 17:11 UT, respectively. It also indicates a decline to minimum particle intensity at 16:55 UT. The mid-latitude region above India has ionospheric irregularity as a result of SEP. As ROTI ≥ 0.5 TECU/min indicates, the equatorial region (Mbarara station) was impacted by the intensification of the SEP during the ascent towards the second peak.

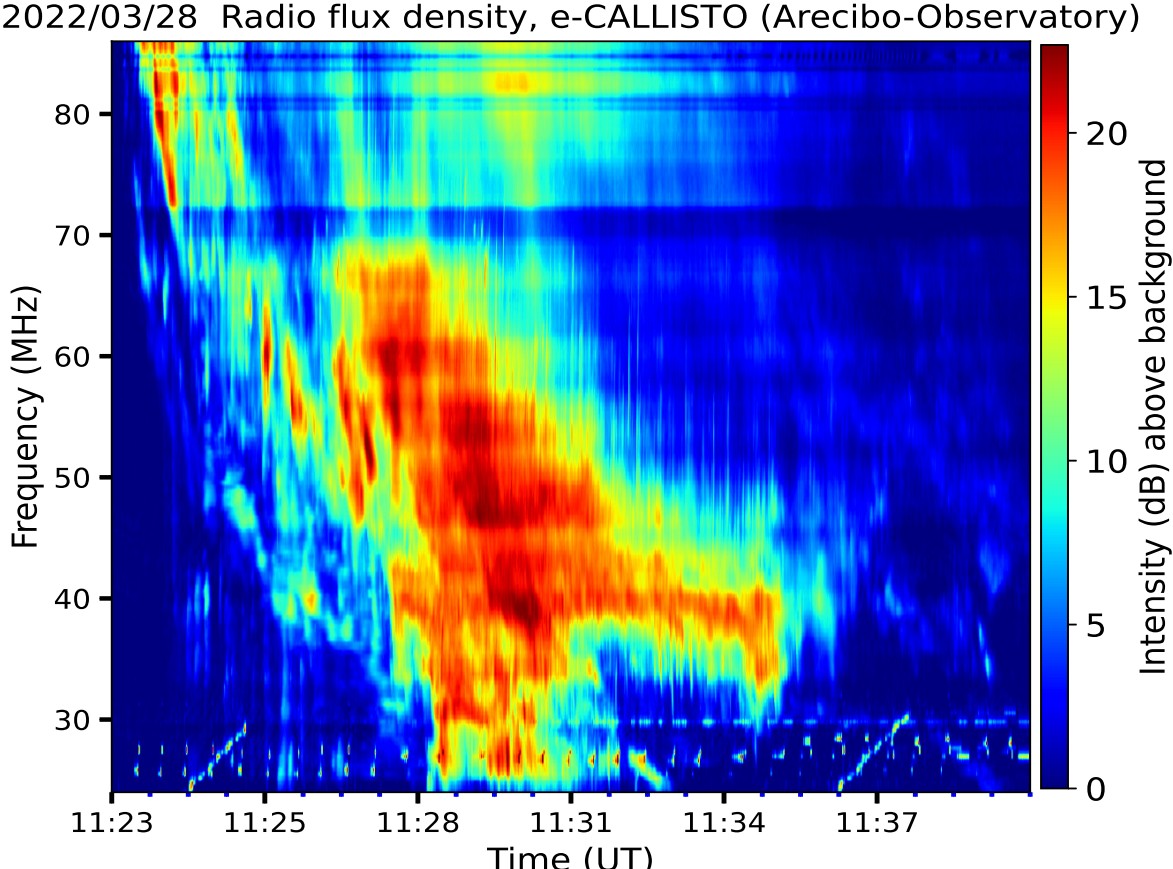

**Figure 9.** The type II radio emissions that are observed on March 28, 2022 from 11:23:12 UT to 11:28:37 UT followed by a type IV radio bursts from 11:26 UT to 11:36 UT.

### 3.2.3 March 31, 2022 event

Type II radio burst observed by e-CALLISTO network at Arecibo Observatory in Puerto Rico, USA from 18:33 to 18:37 UT on 31 March 2022 with 76 - 34 MHz frequency range is overlapped by a type IV radio burst that occurred from 18:36 to 18:41 UT. These bursts are associated with GOES soft X-ray flare M9.6 that started at 18:17 UT, peaked at 18:35 UT and stopped at 18:45 UT from NOAA 12975. These events are associated with halo CME (19:12 UT) with speed of 489 km/s and caused a minor storm on 2 April 2022. A major SEP is also seen on this day, beginning at 03:35 UT and peaking at 04:36 UT. It is outside of the burst time range, and no other bursts were reported to correspond with the SEP (see, https://cdaw.gsfc.nasa.gov/CME_list/daily_plots/sephtx/2022_03/sephtx_20220331.png). This SEP is assumed to be caused by the CIRs (McDonald et al., 1976; Van Hollebeke et al., 1978; Richardson et al., 1993; Tsurutani et al., 2009), with no GOES soft X-ray flare connected with it. Figure 12 displays the daily fluctuation of the ionospheric TEC in terms of ROTI over (a) high

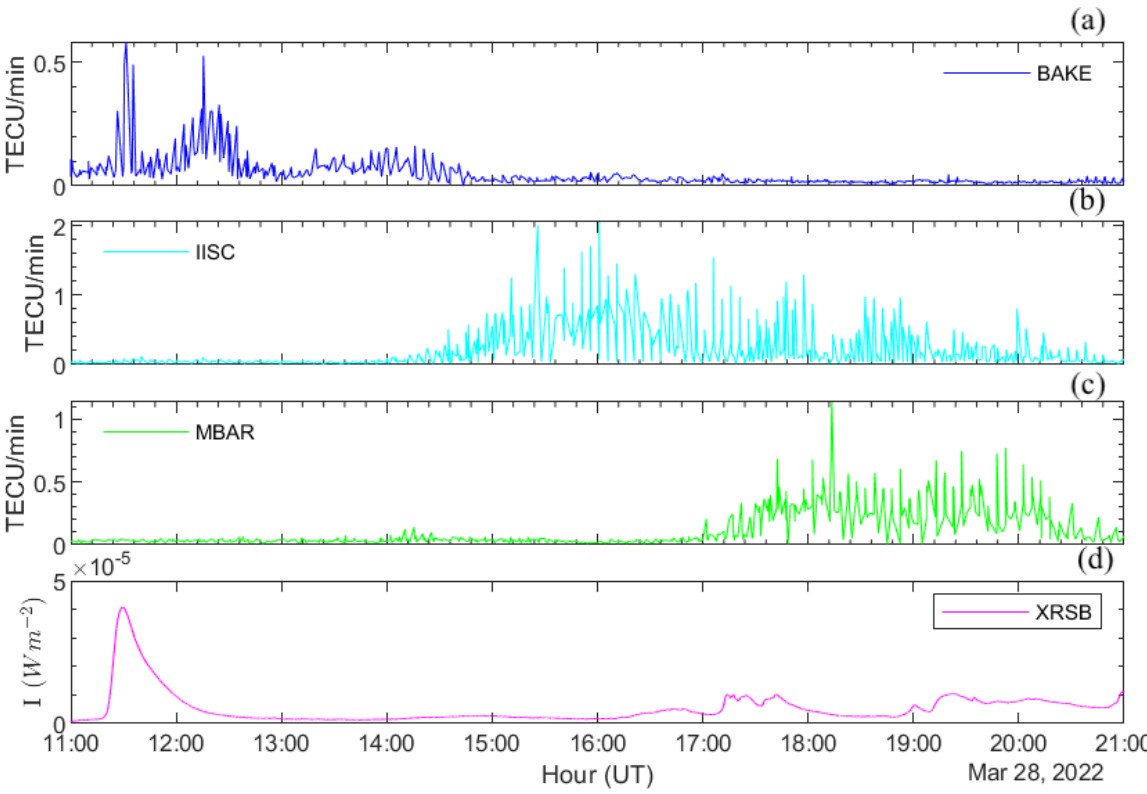

**Figure 10.** ROTI variability in (a) high latitude region, (b) midlatitude region, (c) equatorial region, and (d) associated GOES soft X-ray M4.0 flare.

latitude (INVK), (b) mid-latitude area (IISC), (c) equatorial region (MBAR), and the associated GOES soft X-ray flare flux profile (d). The significant anomalies in high latitudes are thought to be caused by SEP intensification (Figure 12 (a)). However, no notable anomalies have been seen in the mid and equatorial areas as a result of the SEP event. Furthermore, there were no anomalies in any of the locations caused by the M9.6 solar flare. This is believed to be due to electrodynamic coupling of the ionosphere – magnetosphere (Liu et al., 2021; Liu et al., 2021).

### 3.2.4 April 2, 2022 event

The month of April 2022 is also characterized by intense solar activity. On 2 April 2022 between 13:24 and 13:31 UT, type II radio emission is registered within 86 - 30 MHz frequency range followed by type IV radio emission from 13:28 UT to 13:35 UT. They are associated with GOES soft X-ray flare M3.9 that started at 12:56 UT, peaked at 13:55 UT and stopped at 14:44 UT from NOAA 12975. Within the time interval the SEP takes place onset at 14:21 UT and peaks at 15:41 UT. Figure

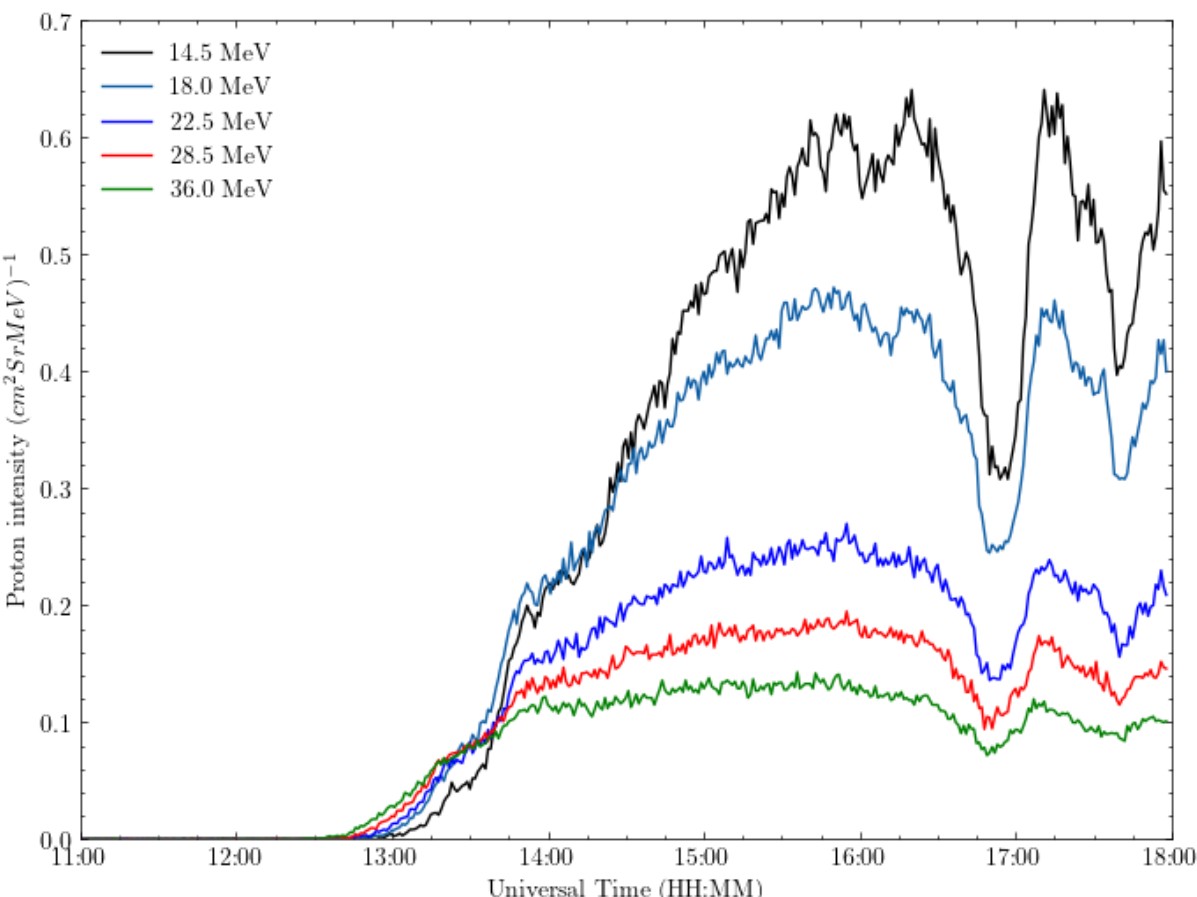

**Figure 11.** Major SEPs in five energy levels occurred on 28 March 2022.

13 illustrates the variability of the ionospheric TEC in response to the the GOES soft X-ray flare and SEP simultaneously over different regions of the globe (a – c). Figure 13 clearly shows no significant ionospheric TEC variation caused by the development of the M3.9 flare as well as from the occurrence of SEP in this time frame over all regions (Figure 13 (a–c)). It should be noted that the particles that are not deflected by the magnetosphere become trapped in the Earth's magnetic field (Oran et al., 2022). Another M4.3 solar flare erupted from the same active region (AR12975) at 17:34 UT, peaked at 17:44 UT, and ended at 17:51 UT. The e-CALLISTO has not detected any radio events, and neither has reported by the space weather prediction centre (SWPC). According to Figure 13, substantial fluctuation of ROTI in the high latitude region (BAKE) began before the second solar flare and is thought to be manifested by the SEP interaction with the magnetosphere. However, the irregularity in ROTI began near the peak of the second flare in mid-latitude (Figure 13 (b)), and it is assumed to be a response to that flare. The equatorial region is unaffected by the three occurrences (Figure 13 (c)). Using the instance scenarios above, it is vital to note that the solar flare lasts between 15 minutes and 2 hours, resulting in continuous ionization throughout the event

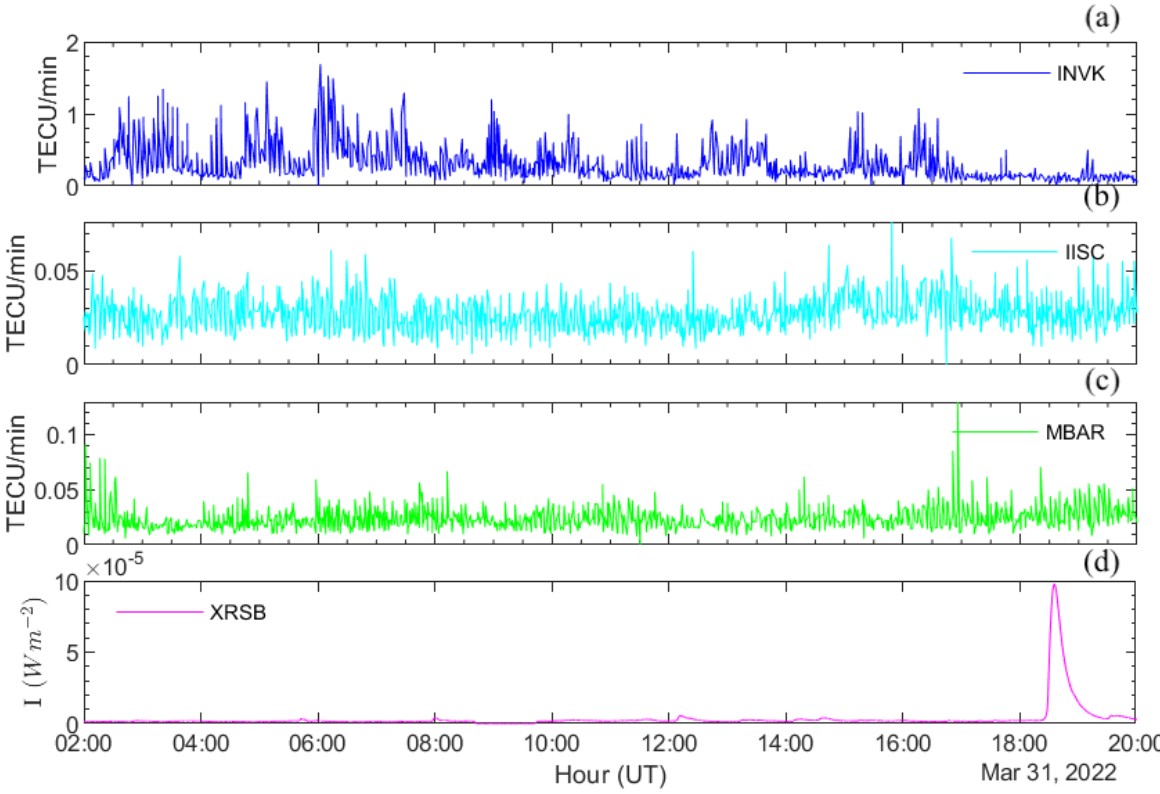

**Figure 12.** Daily variation of the ionospheric TEC in terms of ROTI (a) in high latitude zone (b) in mid latitude region (c) in equatorial region (d) associated GOES soft X-ray flare.

(Tsurutani et al., 2009). Furthermore, the SEPs come quickly after the flare, depending on the particle's kinetic energy, pitch angle, and magnetic connectivity (Hilchenbach et al., 2003; Tsurutani et al., 2009).

## 4 Summary and conclusion

In this study, we report on an analysis of 32 well separated type II radio bursts observed by e-CALLISTO network from May 2021 to December 2022. The parameters of type II radio bursts, such as bandwidth, drift rates and starting frequency are used to derive the corresponding shock parameters: the shock speed, Alfvén speed, Mach number and magnetic field strength. The shock and Alfvén speeds are estimated in the range of 504 - 1282 $\mathrm{kms}^{-1}$ and 368 - 826 $\mathrm{kms}^{-1}$, respectively at heliocentric distance $\sim 1 - 2\ \mathrm{R}_\odot$. The range of measurements that is consistent with the earlier works including the Alfvén speed with 550 - 400 $\mathrm{kms}^{-1}$ given in Cho et al. (2007) at $\sim 1.6 - 2.1\ \mathrm{R}_\odot$. The Alfvén speed of the order of 140 $\mathrm{kms}^{-1}$ to 460 $\mathrm{kms}^{-1}$ at

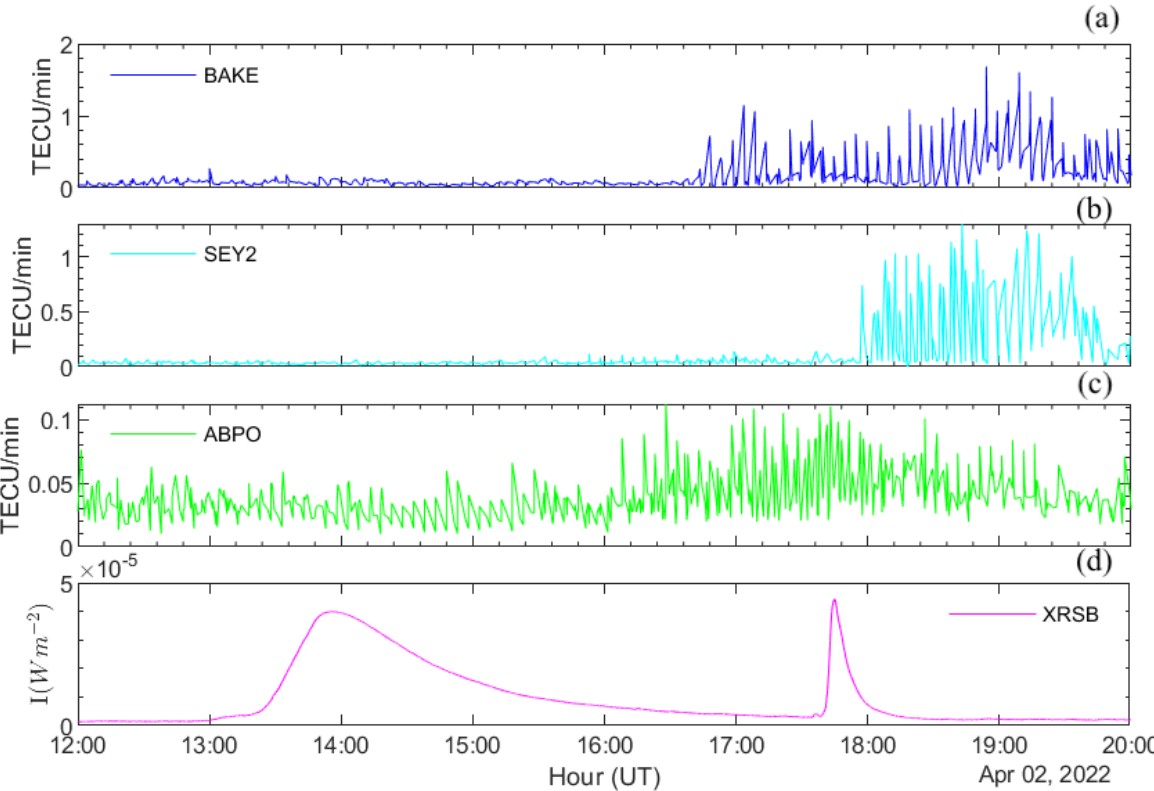

**Figure 13.** Daily variation of the ionospheric TEC in terms of ROTI (a) in the high latitude zone, (b) in the midlatitude region, (c) in the equatorial region, and (d) accompanying GOES soft X-ray flare.

heliocentric distance $\sim 1.2-1.5\,\mathrm{R}_\odot$ reported in Gopalswamy (2011), while Kim et al. (2012) inferred Alfvén speed in the range
of 259 - 982 $\mathrm{kms}^{-1}$ over 3 - 15 $\mathrm{R}_\odot$. The shock speed estimated agrees well with the works of Cunha-Silva et al. (2015) and
Minta et al. (2023) who found shock speed of the order of 200 $\mathrm{kms}^{-1}$ to 810 $\mathrm{kms}^{-1}$. Using Rankine-Hugoniot approximation,
the Mach number of the order of 1.1 to 1.8 is obtained and the magnetic field strength in the range of $\sim 7.8-0.7\,G$ which
is fitted with a single power-law s $B(r) = 6.07 r^{-3.96}\,G$ at the same heliocentric distance. The range of Mach number is in
good agreement with the range of Mach number of $1.59 < \mathrm{M_A} < 2.53$ reported by Mann et al. (2022) and $\mathrm{M_A} \geq 1.5$ by Su
et al. (2022). Our magnetic field strength estimate of the order $\sim 7.8-0.7\,G$ at $\sim 1-2\,\mathrm{R}_\odot$ is well consistent with the work of
Vršnak et al. (2002) who reported the magnetic field strength of 1 - 8 G at $\sim 1.6\,\mathrm{R}\odot$ and also with 6 - 5 G at $\sim 1.5-1.7\,\mathrm{R}_\odot$
found in Ramesh et al. (2010). According to the current research, 19 of the 32 type II radio events are precursors for space
weather because they are connected to immediate space weather phenomena like radio blackouts and polar cap absorption
events, exhibit band-splitting characteristics, or are followed by type III and IV bursts. The current study's findings also reveal

that ionospheric disturbances are common depending on the strength of flare classes and/or SEPs, as evidenced by ROTI irregularities, and solar radio type II observations are used as indicators in this situation. This article demonstrates that because type II bursts are connected to space weather hazards, understanding various physical properties of type II bursts aids in the prediction and forecast of space weather.

*Author contributions.* T. Ndacyayisenga, Jean Uwamahoro, J. C Uwamahoro and D. Okoh conceived the presented idea and the design of the
315 study. T. Ndacyayisenga manually gathered the data used. C. Monstein and D. Okoh, helped in programming for data analysis. Analysis and intrepretation of the results are done by T Ndacyayisenga who later drafted the manuscript. This manuscript is reviewed by Jean Uwamahoro, J. C Uwamahoro, Rabiu Babatunde, D. Okoh, K. Sasikumar Raja and C. Kwisanga.

*Competing interests.* The authors declare that the research was conducted in the absence of any commercial or financial relationships that could be created as a potential conflict of interest.

*Acknowledgements.* This work was supported by International Science Programme (ISP) through Rwanda Astrophysics, Space and Climate Science Research Group (RASCSRG) and Centre for atmospheric Research through National Space Research and Development Agency, Abuja, Nigeria. We thank FHNW, Institute for Data Science in Brugg/Windisch, Switzerland for hosting the e-Callisto network and the individual Callisto operators such as University of Alaska Fairbanks - Geophysical Institute, Arecibo Observatory and Astronomical Society of South Australia . The authors also thank the providers of all the data used from SOHO/LASCO; NOAA; GOES, SWPC, African Geodetic
Reference Frame (http://afrefdata.org), Solar monitor (https://solarmonitor.org/), Coordinated Data Analysis Web (https://cdaweb.gsfc.nasa.gov/), and the UNAVCO Archive of GNSS Data (ftp://data-out.unavco.org). Finally, we would like to thank the referee for their useful comments and suggestions during the review of this manuscript.

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
