# Peer review of "Low-frequency solar radio type II bursts and their association with space weather events during ascending phase of solar cycle 25"

_EGUsphere, 2023_

## Author Comment (AC1)

*Firstly, we thank the referee for providing the useful comments on our manuscript. Following the referee's comments, we will carefully go through the manuscript and revise it. Herewith, we provide the answers to the referee's comments:*

**Answers to the Referee Comments # RC1**

The authors analyzed 31 type II bursts from e-CALLISTO observations during the period of May 2021 to December 2022 (the ascending phase of the current solar cycle). Based on measurements of the dynamic spectra, they estimated the physical parameters of the associated shocks. They also examined associated X-ray flares and CMEs. The authors further examined in detail space whether effects of bursts on October 9, 2021, and during March 24 - April 3, 2022. They conclude that 15 of their events were associated with space weather events.

Their results are new and interesting, hence the article merits publication after some improvements:

**1. The authors should provide a table with the statistics of their results (range of values, average, rms), together with values from previous works for easy comparison.**

Answer: Figure 3 gives the statistics required and the comparison with previous was done in the text during discussion.

**2. Lines 107-109: please provide the values of the slope derived by Vršnak et al., 2002, and Umuhire et al., 2021.**

Answer: The correlation coefficients are CC=0.93 and CC-0.83 for Vršnak et al., 2002, and Umuhire et al., 2021, respectively.

**3. The difference between the CME speed derived from the dynamic spectra and that of LASCO should be discussed further.**

Answer: The difference is discussed in the revised version: LASCO Coronagraphs observe the Sun from 1.1 to 3, 2.5 to 6, and 4 to 32 solar radii (Rs) in C1, C2, and C3, respectively. LASCO-C2, which observes between 2.5 and 6 Rs, was used in

our study. The difference in CME speed from dynamic spectra and LASCO is attributed to the CME's central position angle as observed by LASCO, implying that the shock weakened and dissipated before entering LASCO's field of view (FOV) (Gopalswamy et al., 2012, ApJ, 744,72).

**4. (i) Figure 4: (a) Please label the axes; (b) Add your own measurements. (c) In the insert, replace equations with the model names.**

Answer: (a) The axes labels will be added. (b) Our own measurements are given by a dashed curve and (c) the equations will be replaced by their corresponding model names.

**(ii) Where does the "quiet Sun magnetic field model" come from?**
**Answer**: It is found in Gopalswamy et al., 2001. J. Geophys. Res., 106, 25261.

**5. Lines 200-220: Can you identify which TEC enhancements are associated with which type II?**

Answer: Because of their association with solar phenomena, type II bursts aid in tracking disturbed days. If there is a significant increase in TEC on a specific day and a type II radio burst registration on that same day, it is proof that the two are linked. We did not reinstate type II bursts because the days for TEC calculations were chosen based on the occurrence of type II bursts.

**6. The authors conclude that 15 of their events were associated with space weather events. This is an important result and requires further discussion, preferably in a separate section. For example, what were the differences between these 15 and the other type IIs? Are there any observable type II characteristics that enhance the probability of space weather effects?**

**Answer:** Solar radio bursts are electromagnetic radiation that travels at the speed of light. They are used as a proxy to provide early warning of upcoming events. According to a recent study by Chernov and Fomichev, 2021 (ApJ,922,82), the registration of type II radio bursts is a signature of shock acceleration in the solar corona. As a result, by observing type II radio bursts, it is possible to trace out

associated solar phenomena such as solar flares and coronal mass ejections, which are the drivers of space weather. Furthermore, those 15 type II bursts are all associated with the aforementioned space weather driver and either proton enhancement due to flare explosions or polar cap absorption events. Finally, it is affected by the location of associated active regions, whether they are associated with M or X classes of flares and the strength and direction of associated CMEs.

Finally, the authors may find interesting the review, doi:10.3389/fspas.2020.591075 on radio measurements of the magnetic field.

---

## Author Comment (AC2)

*Firstly, we thank the referee for providing the useful comments on our manuscript. Following the referee's comments, we will carefully go through the manuscript and revise it. Herewith, we provide the answers to the referee's comments:*

**Answers to the Referee Comments # RC1**

The authors analyzed 31 type II bursts from e-CALLISTO observations during the period of May 2021 to December 2022 (the ascending phase of the current solar cycle). Based on measurements of the dynamic spectra, they estimated the physical parameters of the associated shocks. They also examined associated X-ray flares and CMEs. The authors further examined in detail space wheather effects of bursts on October 9, 2021, and during March 24 - April 3, 2022. They conclude that 15 of their events were associated with space weather events.

Their results are new and interesting, hence the article merits publication after some improvements:

**1. The authors should provide a table with the statistics of their results (range of values, average, rms), together with values from previous works for easy comparison.**

Answer: A table of statistics with comparisons of previous works that have analyzed more than two events.

| Epoch | # events | Mean shock speed (km/s) | Mean Alfven speed (km/s) | B-field (G) | Height range (solar radius) | Authors |
|-------|----------|-------------------------|--------------------------|-------------|-----------------------------|---------|
| 2021 - 2022 | 31 | 893 | 604 | 8.0 - 0.6 | 1.0 - 2.0 | This work |
| 2013 - 2014 | 4 | 843 | - | 1.8 - 1.3 | 1.68 - 1.91 | Kishore et al., 2016 |
| 1996 - 2007 | 10 | 1288 | 555 | 0.105 - 0.006 | 3 - 15 | Kim et al., 2012 |

**2. Lines 107-109: please provide the values of the slope derived by Vršnak et al., 2002, and Umuhire et al., 2021.**

Answer: The slopes are $\epsilon = 1.89$ and $\epsilon = 1.33$ for Vršnak et al., 2002, and Umuhire et al., 2021, respectively.

**3. The difference between the CME speed derived from the dynamic spectra and that of LASCO should be discussed further. I suppose you imply that the shock decelerates as it moves up in the corona. This should be included in the text, together with examples from the literature.**

Answer: The difference in CME speed between dynamic spectra and LASCO is attributed to the CME's central position angle as observed by LASCO, implying that the shock may be weakened and dissipated before entering LASCO's field of view (FOV) (Gopalswamy et al., 2012, ApJ, 744,72). On the other hand, the shock decelerates in the case of a decline in its intensity or when it breaks. Furthermore, the shock can accelerate particles (electrons, protons, and heavy ions) if they become supercritical ($MA > 1$) (Kennel et al., 1985).The type II burst only serves as a time marker for when the shock occurs. It should be noted that type II radio emission can come from anywhere on the shock front: the nose or the flanks, depending on which location is best for electron acceleration (Gopalswamy et al., 2013. Ad. Spac. Resc, 51, 1981-1989).

**4. (i) Figure 4: (a) Please label the axes; (b) Add your own measurements. (c) In the insert, replace equations with the model names.**

Answer: (a) The axes labels are added. (b) Our own measurements are given  by magenta starts and fitted with a dashed curve and (c)  the equations are replaced by their corresponding model

names.

[Figure]

**(ii) Where does the "quiet Sun magnetic field model" come from?**
**Answer**: It is found in Gopalswamy et al., 2001. J. Geophys. Res., 106, 25261.

**5. Lines 200-220: Can you identify which TEC enhancements are associated with which type II?** **You should clarify your criteria of association and, based on these, give the burst numbers (from your table 1) that you can positively associate with space weather effects.**

Answer: Because of their association with solar phenomena, type II bursts aid in tracking disturbed days. The TEC was analyzed on 25 type II radio bursts, which are associated with both solar flares and CMEs, by selecting GNSS stations in equatorial, mid-latitude and high-latitude regions. Particularly, the TEC enhancements of October 8 - 11, 2021, and that of March 24 - April 3, 2022, are described in more detail in the manuscript.

**6. The authors conclude that 15 of their events were associated with space weather events. This is an important result and requires further discussion, preferably in a separate section. For example, what were the differences between these 15 and the other type IIs?**

**Are there any observable type II characteristics that enhance the probability of space weather effects? If not, you should still state that you could find no differences between the two type II groups.**

**Answer: In the present study, 15 of the 31 type II radio events were found to be related to space weather phenomena, including radio blackouts or polar cap absorption events brought on by solar proton enhancement and solar energetic particle phenomena.** However, out of the 31 type II radio events, 18 are diagnostic of space weather, 10 of which have band-splitting characteristics, and the other 8 are preceded by type III bursts or followed by type IV radio bursts.

---

## Author Comment (AC3)

*Firstly, we thank the referee for providing useful comments on our manuscript. Following the referee's comments, we will carefully go through the manuscript and revise it. Herewith, we provide the answers to the referee's comments:*

**Answers to the **Anonymous Referee #2**

The paper presents observations of type II bursts during the ascending phase of SC25 achieved with e-CALISTO instruments as well as some associations of these type II bursts with space weather features. The paper presents potentially interesting events but needs some major improvements before publication. The conclusions on the type II analysis should be more emphasized (**what is new with these observations with respect to previous observations?**) and the link with the space weather effects should be investigated in more details (**the fact that effects are seen at the same time as the type II burst does not explain the physical link between both observations**).

**Answer:**

The current study reports on the first observation of type II solar radio bursts using ground data in solar cycle 25 with the emphasize of diagnosing the status of the progress of the solar activity. Type II radio observations are among the quick indicators of the solar activity as they provide clues to diagnose space weather hazards such as geomagnetic storms and radiation effects. It is still difficult to predict space weather phenomena on a practical basis, but continuous monitoring of solar radio bursts plays an important role due to their origin and characteristics of being associated with space weather drivers. The ground observation of type II bursts (in metric) with large ground coverage contributes to early warning of solar activity status for associated geoeffective CMEs erupting in the corona and interplanetary medium.

**Here are some detailed comments and questions:**

Abstract

**Line 3 : The authors mention solar storm disturbances, but they should precise which kind of disturbances since in the following of the paper they mention TEC enhancement, radio blackouts, polar cap absorption, etc.**

**Answer:** The authors intended to refer to geomagnetic storms and subsequent effects in the magnetosphere and ionosphere  such as radiation storms, when they used the term "solar storm disturbances." Therefore, the line is modified as follows: Being electromagnetic radiation that travels at the speed of light, type II radio bursts can serve as  proxy to provide early alerts of incoming solar storm disturbances  such as geomagnetic storms and radiation storms, which may lead to ionospheric effects.

**Line 12: The authors mention solar proton enhancements and solar particle events. What is the difference ?**

**Answer:** There is no difference between the two in the context of the current work. As a result, solar proton enhancement will be retained while the other is eliminated.

Observations
- **In the section, "Derivation of shock characteristic parameters', the authors quote the papers by Vrsnak et al. (2001, 2002) to estimate the density jump across the shock. However, in these papers, the BDW used in equation 2 refers to the instantaneous band splitting of the type II emission and not to the instantaneous bandwidth mentioned in equation 1. The authors should give clearer explanations of the description of the observations and the use of the relations derived from these papers. Do they observe band-splitting for all the type II bursts they analyzed? They should also correct the wording in section 2 as well as Figure 1's caption.**

**Answer:** The band-splitting of type II bursts is an important feature for calculating the coronal magnetic field. The bandwidth (BDW) is the width of the fundamental or harmonic band caused by the presence of a band-splitting type II burst. Cho et al. (2007) (ApJ,665:799), for example,

used the width measured on the fundamental band of a band-splitting type II burst. Because all of the type II bursts examined lacked the band-splitting feature, we linked the width of the fundamental band to the ambient density jump to ensure consistency in computation; otherwise, we should have chosen only band-splitting type II bursts. However, the paraphrasing of Section 2 will be done in the revised version.

- **The authors assume a density variation as r-6.13 as used in Gopalswamy (2011). Given that the different e-CALLISTO instruments observe in different frequency bands (i.e. radio emission produced at different coronal heights), is this choice of density model relevant for all the events?**

**Answer:** The density model chosen is applicable to all type II bursts studied because it describes the variation within 1 - 3 solar radii (Rs) coronal height, and all of our events fall within that range.

- **In equation 6, the authors should indicate the units.**

**Answer:** Equation 6 is as follows: $B(G) = 5.1 \times 10^{-5} f_l(MHz) V_A(km/s) G$

**- The end of section 2.2 contains information on GPS data and is not relevant to the derivation of shock characteristic parameters. A new section should be created for the discussion of the GPS data.**

**Answer:** Section 2.3 is created for this part.

GPS data from ground-based GPS receiver stations around the world were used to analyze the ionospheric total electron content (TEC) for disturbed days identified by type II radio burst observations in this study. As GPS data are usually provided in Receiver Independent EXchange (RINEX) format, TEC were derived from Rinex files using the GPS TEC software developed at Boston college, assuming a thin shell ionosphere at the altitude of 350 km. Details on the software used to derive TEC are provided in Seemala & Valladares, 2011; Uwamahoro et al., 2018, and references therein.

1. Seemala, G., & Valladares, C. (2011). Statistics of total electron content depletions observed over the South American continent for the year 2008. Radio Science, 46, RS5019. https://doi.org/10.1029/2011RS004722.

2. Uwamahoro, J. C., Giday, N. M., Habarulema, J. B., Katamzi-Joseph, Z. T., & Seemala, G. K.(2018). Reconstruction of storm-time total electron content using ionospheric tomography and artificial neural networks: A comparative study over the African region. Radio Science, 53. https://doi.org/10.1029/2017RS006499

**Results and discussions**

(a) **The e-CALLISTO network consists of many stations working in different frequency bands. The authors should specify in table 1 the name of the instrument(s) which observed the different type II bursts. Only a small number of the listed type II bursts starts at frequencies above 100 MHz. This may be linked to the instrument or reveal different characteristics of type II bursts. The authors should also discuss how the CME parameters are derived.**

**Answer:** Given its size, Table 1 might become overloaded with additional data. As a result, we can make a new table that lists the instruments that were used, their locations, and their frequency ranges.

| S N | File ID | Country | Lat($^0$) | Long($^0$) | Obs. Frequency Range (MHz) | # of events |
|-----|---------|---------|-----------|------------|----------------------------|-------------|
| 1 | Australia_ASSA | South Australia | -30.00 | 136.21 | 15 - 87 | 11 |
| 2 | Arecibo_Observatory | Puerto Rico, USA | 18.22 | -66.59 | 15 - 87 | 9 |
| 3 | GREENLAND | Greenland | 67.00 | -50.72 | 10 - 110 | 3 |
| 4 | ALASKA_HAARP | ALASKA | 64.84 | -147.72 | 5 - 87 | 2 |
| 5 | ALMATY | Kazakhstan | 43.22 | 76.83 | 45 - 165 | 1 |

| 6 | BIR | Ireland | 16.61 | 77.51 | 10 - 100 | 1 |
| 7 | INDIAN_OOTY | India | 11.41 | 76.69 | 45 - 165 | 1 |
| 8 | KASI | South Korea | 36.35 | 127.38 | 150 - 400 | 1 |
| 9 | MEXICO_LANCE | MEXICO | 19.81 | -101.69 | 50 - 90 | 1 |
| 10 | SWISS-Landschlacht | Switzerland | 47.63 | 9.25 | 15 - 87 | 1 |

We compared the values of the derived shock parameters (shock speeds and Alfven speeds) with the speeds of the CME parameters, which were taken from the catalog.

(b) Line 102: The authors use a relationship published by Gopalswamy et al (2013) to derive the shock formation height of type II. They should discuss how this relationship was found and whether it can be used on another sample of data (like the present one).

Answer: According to Gopalswamy et al. (2013), the correlation between the starting frequencies of type II radio bursts and CME heights is best fit by a power-law: $f(r) = 307.87r^{-3.78} - 0.14$. It was established for a sample of 32 metric type II bursts at 1.20–1.93 solar radii (Rs). Umuhire et al., 2021 (Sol. Phys. 296:27), used this relation on a sample of 40 metric type II bursts at 1.16–1.90 Rs. As a result, we used this relationship to estimate the shock formation height for our sample of 31 metric type II bursts.

(c) Figure 2: Most of the type II bursts have starting frequencies below 100 MHz. Is the correlation coefficient different if only type II bursts with starting frequencies below 100 MHz are considered ? Same questions with the relationship between the frequency drift rate and the starting frequency ?

Answer: When only type II radio bursts with starting frequencies less than 100 MHz are considered, a very weak correlation (CC = 0.522) between frequency drift rates and frequencies is obtained

because the observation is dominated by points with nearly equal values. This also has an impact on their relationship. However, considering type II with starting frequencies less than 200 MHz yields a different correlation coefficient (CC = 0.845), which is still a good correlation between the two parameters, and the relationship between the frequency drift rates and the frequencies becomes:

$|\frac{df}{dt}| = 0.001f^{1.05}$ for 30 events out of 31. Therefore, figure 3 is modified as follows

[Figure]

**(d) Figure 3: It would be interesting to plot v derived from the dynamic spectrum with respect to v derived from the CME.**

**Answer:** Figure 3 is replaced by the following.

[Figure]

**Section 3.2 : Associated Space Weather implication**

**(e) Figure 5 bottom : is this plot showing a prediction or real observations? Is this HF absorption linked to the arrival of protons or to the ionizing flux from the flare ? How can the type II burst observations explain this HF absorption ?**

**Answer:** (i) The observation of the bottom of Figure 5 is real, that is why some of the type II bursts are associated with immediate effects.

(ii) The HF absorption is linked to the ionizing flux from the flare. Typically, the amount of ionizing radiation is measured in terms of the flux of particles or photons per unit area per unit time.

(iii) Type II bursts act as an alarm for space agencies to track incoming solar events. From Figure 5, the absorption was recorded later than type II, so type II is a proxy.

**(f) Correct the time of the type IV burst time in line 179 as well as in the caption of figure 7**

**Answer:** The type IV burst time is now 11:26 UT to 11:36 UT.

**(g) Similar question for figure 8 as the one asked for figure 5. Are the times the same for figure 8 top and bottom?**

**Answer:** The top and bottom times of Figure 8 differ. The bottom image shows a polar cap absorption event (protons ejected towards the pole), while the top image shows a radio blackout (an increase in proton fluxes across the entire continent of Africa caused by a flare). The text makes the distinction, and the images are included here with the dates and times of their records.

[Figure]

[Figure]

**(h) There are a lot of acronyms from lines 187 to 195 (SPE,SEP,PCAE). Please precise their meaning.**

**Answer**: The acronyms stand for solar proton event (SPE), solar energetic particles (SEPs), and polar cap absorption event (PCAE) or polar cap absorption (PCA) event. However, the SEPs will be removed for the sake of consistency of the current work.

**(i) Figure 9 : The authors should precise the link between the TEC enhancements in the different stations and the many type II bursts reported during this period.**

**Answer:** In the current study, type II solar radio bursts were used as selection criteria for disturbed days due to their connection to solar phenomena (e.g., radio blackouts). By choosing GNSS stations in equatorial, mid-latitude, and high-latitude regions of the affected areas, the TEC was examined on 25 type II radio bursts, which are linked to both solar flares and CMEs.

In particular, the TEC enhancements of 24 March – 1 April 2022 are described in Figure below (New Figure 9 due to data gap), where four type II radio bursts were observed in the aforementioned range (as listed in Table 1 of the manuscript). The four bursts are associated

with CMEs of mean speed of 691 km/s and estimated shock speed of 990 km/s. However, no CME has reached the magnetosphere because no geomagnetic storm in the selected interval. With the help of the solar monitor website (https://www.solarmonitor.org), there is a presence of large coronal holes and one can expect a corotating interaction, as a result of substorms on 26, 27 of March 2022, and 1 April 2022 (**panel e of figure below**). Using the line plots, Figure…. shows the diurnal variation of TEC over four different stations (**mbar**: Mbarara, **abpo**: Madagascar, **falk**: Falkland Islands and **bogt**: Bogota). Because the radio bursts are associated with radio blackouts, the stations were selected in the affected areas to ensure strong diurnal variation.

**Panel (a)**: The trend of TEC variation shows a decrease of 10 TECU on 26 March 2022, compared to normal TEC (normal maximum TEC=65 TECU), and an increase of 8 TECU on 31 March 2022. The solar flare is responsible for the decrease in TEC on March 26, 2022, and CIR (Kp=5) is the cause of the increase in TEC on 31 March 2022.

[Figure]

**Panel (b)**: This GNSS station experienced 5 TECU drop on 26 March 2022 and TEC enhancements of 8 and 12 TECU on 29 March and 31 March 2022, respectively. Such drop and

enhancements are caused by solar flares and CIR (Kp=5), respectively. **Panel (c)**: TEC is enhanced on 31 March 2022 compared to other days. **Panel (d)**: The diurnal variation of TEC is increased by more than 24 TECU on a daily basis on March 25, 26, 30, 31, and April 1, 2022, due to CIR (kp=4), solar flare, and CIR (Kp=5), respectively, over Bogota station. The variations of TEC over these stations are attributed to the ionizing flux from flares and CIR. The diurnal variation is prominent at all stations, which corresponds to the same feature plotted on the contour maps by taking the entire range at each station.

**(j) Last part of section 3.2 : the authors claim that 15/31 events have immediate space weather effects but this is not really shown in the paper. More generally, the discussion between the type II observations and the space weather effects is vague. It is not clear how the observations of type II bursts can predict TEC enhancements since they may be due to the flare ionized flux or to the arrival of energetic particles. Furthermore, it is not clear why some type II bursts sare associated with space weather effects and why others are not.**

**Answer:** In the current study, 15 of the 31 type II radio events were linked to immediate space weather phenomena, such as radio blackouts or polar cap absorption events caused by the solar proton enhancement phenomenon. However, 18 of the 31 type II radio events are diagnostic of space weather, 10 of which have band-splitting characteristics, and the remaining 8 are preceded or followed by type III or IV radio bursts.

The prediction of TEC necessitates the development of a model, which is beyond the scope of this work. We only used type II radio bursts to select disturbed days and analyze their TEC to see the behaviour in comparison to the days where we did not have any type II bursts. Therefore, given that Type II bursts are EM, travelling at speed of light , continuous monitoring / observations and record of their data is useful and can be used in various models to predict TEC during disturbed days. The TEC modelling is out of scope of the current paper.

---

## Referee Report (RR1)

The authors have largely improved the paper since the last version, in particular the section on type II bursts. The section 3 on associated space weather implication still needs major improvement and clarification.

Comments on section 3 :

The authors use TEC information from different GNSS stations to study the implication of type II bursts on space weather effects. They consider two periods to analyse these effects. They should introduce in this section the effects on TEC which are expected as a response to a flare (e.g. linked to the radio black-out phenomena mentioned earlier) or to particles. What are the time scales for the effects on TEC ? How are these effects varying with terrestrial longitude/latitude ? The figures showing the TEC evolution should be better explained so that the reader clearly unerstands what is usual diurnal variation and what is due the effect of the flares . The authors also mention several times in this section CIR. These phenomena are not described ealier in the paper and I wonder what is the link with the type II/shock/CME.

Additional comment on the abstract (lines 11 to 14) :

> **« The current study finds that 18/31 type II radio events are precursors for space weather because they are associated with immediate space weather events such as radio blackouts and polar cap absorption events and exhibit band-splitting features or are followed by type III and IV bursts »**

> This sentence should be split in two. As it is written, it can be understood that type II radio events are precursors for space weather because they exhibit band-splitting features, which is not the case. Also are they followed by or associated with type IV bursts. (This sentence appears in a more or less similar way elsewhere in the paper).

---

## Author Response (AR3)

**Detailed Answer to the comments**

**Firstly, we thank the referee for providing useful comments on our manuscript. Following the referee's comments, we will carefully go through the manuscript and revise it. Herewith, we provide the answers to the referee's comments:**

Abstract:

**« The current study finds that 18/31 type II radio events are precursors for space weather because they are associated with immediate space weather events such as radio blackouts and polar cap absorption events and exhibit band-splitting features or are followed by type III and IV bursts »**

This sentence should be split in two. As it is written, it can be understood that type II radio events are precursors for space weather because they exhibit band-splitting features, which is not the case. Also are they followed by or associated with type IV bursts. (This sentence appears in a more or less similar way elsewhere in the paper).

Answer:

The statement is rephrased as follows, and everywhere in the manuscript is harmonized.

Based on the current analysis, it is found that 18 out of 31 type II bursts are associated with immediate space weather events in terms of radio blackouts and polar cap absorption events, making them strong indications of space weather disruption. Consistent with previous research, type II radio bursts, that often may occur in association with type III and type IV bursts, are probably the most relevant events to predict the space weather.

**Comments on section 3**

1. The authors use TEC information from different GNSS stations to study the implication of type II bursts on space weather effects. They consider two periods to analyze these effects. They should introduce in this section the effects on TEC which are expected as a response to a flare (e.g., linked to the radio black-out phenomena mentioned earlier) or to particles.

   **Answer:** Total electron content (TEC) is one of the significant ionospheric parameters to indicate the integrated electron density along the line of sight/path of the signal. Solar flare radiation, mainly from the extreme ultraviolet and X-ray wavelengths, interacts with ionospheric constituents, causing an immediate rise in total electron density in the ionosphere. It is known that ionospheric TEC varies day to day, season, latitude, and longitude, along with the solar and geomagnetic activity (Seemala et al 2023).

   The extent of ionospheric TEC augmentation appears to be determined by the kind of solar flares (Liu et al, 2006, J Geophys Res, 111; Kumar and Singh, 2012, IJR & Space Physics, 141). During the peak of an X-ray solar flare, ionospheric TEC abnormalities are frequently suppressed due to accelerated solar energetic particles (Oljira, 2023, AdSp Res, 3868). The rate at which TEC varies temporally is related to the effective flare radiation flux (Wan et al 2002, Sci. China Ser A, 142).

2. How are these effects varying with terrestrial longitude/latitude ?

   Answer: All terrestrial regions are exposed to such effects depending on the strength of radiation because weak particles are sent to polar regions (high latitude) where they give rise to aurorae phenomena.

3. The figures showing the TEC evolution should be better explained so that the reader clearly understands what is usual diurnal variation and what is due the effect of the flares .

Answer:

For the sake of clarifying the abnormalities in TEC in the revised manuscript, the Rate of TEC Index (ROTI) is now used to quantify the variability of TEC in line with observed solar events. In this regard, the 5 -minute averaged ROTI data over the stations listed in Figure below are plotted as a function of universal time (UT).

[Figure]

It is important to note that three events, among others, are associated with major solar energetic particles (> 10 MeV), and the illustrative examples given in the current documents are subjected to the analysis of their effects on the ionosphere.

For example, on 28 March 2022, a type II burst observed by the CALLISTO spectrometer at the Arecibo Observatory during 11:23 UT - 11:28 UT, followed by a type IV radio burst during 11:26 UT - 11:36 UT are associated with an M-class flare that erupted at 10:58 UT and continued until 11:45 UT.

The following figures show the solar flare and major SEP associated with the above solar radio bursts. The left figure shows the solar flare while the right shows the SEP in 5 energy channels.

[Figure]

The following two figures show the ROTI variation in response to the solar flare and SEP. At high latitude at the BAKE (Canada) station, the ROTI reached 0.37 TECU/min at the peak of flare at 11:29 UT and decreased to 0.08 TECU near the end of solar

flare. However, no TEC variation was detected in the equation or mid-latitude regions. Using one station in the equatorial region (MBAR, Uganda), there is prominent TEC enhancement (ROTI>0.2 TECU/min) in response to the SEP at to its peak time (17:52)

[Figure]

[Figure]

4. What are the time scales for the effects on TEC ?

   Answer: The effects are proportional to the time rate of change of flare radiations (Liu et al, 2006). Further note that ionospheric TEC varies day to day, season, latitude and longitude along with the solar and geomagnetic activity (Seemala et al 2023).

5. The authors also mention several times in this section CIR. These phenomena are not described earlier in the paper.

   Answer:

A statement explaining CIR is added to the revised manuscript as follows.

Corotating Interaction Region (CIR) is a compressed solar wind plasma that forms between slow solar wind and high speed solar wind streamers (HSS) that emanate from the coronal holes (Belcher and Davis, 1971; Siscoe, 1972; Krieger, Timothy, and Roelof, 1973). During the solar minimum, the corotating interaction regions (CIRs) are the principal source of energetic particles in the heliosphere (e.g., Richardson et al., 1993). CIRs develop when a rapid solar wind emerges from a coronal hole that reaches low latitudes and overtakes a parcel of slow solar wind generated by the Sun at earlier times. The solar rotation causes these plasmas of different speeds to become radially aligned and interact (e.g., Gosling and Pizzo, 1999).

6. I wonder what is the link with the type II/shock/CME.

Answer:

Coronal mass ejection is a huge bubble of plasma that is ejected from the lower corona and propagates through the heliosphere. If the speed of CME exceeds the local background Alfvén speed, it produces a plasma shock at the CME nose or at the flanks. Such shocks further produce the radio type II bursts via plasma emission mechanism. Note that shocks are observed at extreme ultraviolet, white lights and radio wavelengths (Maguire et al, 2020, Carley et al 2021). Coronal Mass Ejections trigger space weather hazards by compressing the Earth's magnetosphere upon their arrival at the Earth which results in channelizing the particles into the Earth atmosphere to produce Auroras. CMEs are also responsible for geomagnetic storms, power grid disruptions, and to accelerate solar energetic particle (SEP) events etc. Since type II bursts are signatures of the coronal shocks and reach Earth in 8 minutes, they serve as a proxy to predict space weather events.

---

## Author Response (AR4)

**Answers to the comments**

**Firstly, we thank the referee for providing useful comments on our manuscript. Following the referee's comments, we have carefully gone through the manuscript and revised it. Herewith, we provide the answers to the referee's comments:**

Looking at the manuscript I identified the following additional points that the authors should look into before the publication

Page 2: "The ROTI is calculated by first describing the fluctuation in plasma caused by ionospheric disturbances using ROT (e.g., Pi et al., 1997; Cherniak et al., 2014; Liu et al., 2019). The ROTI is used to visualize the strength ...|"

From the text, an additional "second" (after the first step) and possibly further steps seem to be missing. Could the author add those steps and describe in more detail how the index is calculated (ideally by also providing a formula) and what the difference between ROT (line 38) and ROTI (line 35). I would recommend a dedicated subsection (not the introduction) to introduce ROTI in more detail, as it seems an important parameter of the paper.

**Answer**:
ROT stands for Rate Of change of TEC while ROTI stands for Rate of TEC Index.
The paragraph is paraphrased as follows:
Pi et al. (1997) developed an index known as the Rate Of change of TEC (ROT), that is based on the time rate of various phase changes in dual-frequency GNSS signals crossing the same ionospheric parcel and is expressed in TECU/min (1 T ECU = $10^{16}$ electrons/m$^2$). Depending on the dual-frequency GPS signals, ROT explains the irregularities on various length scales. The standard deviation of the ROT is used to construct the Rate Of TEC Index (ROTI) which has the same unit as ROT (e.g., Pi et al., 1997; Cherniak et al., 2014; Liu et al., 2019). ROTI describes the small -scale irregularity of the line of sight electron content resulting from the ionosphere (Pi et al., 1997; Liu et al., 2019).

However, the subsection to introduce ROT & ROTI in more detail is not necessary because the above paragraph explains the meaning of the two indices and their important roles, while the formulas for their calculations are given in Section 2.3, Equations 7 and 8, respectively.

Page 3: The sentence "The latter is accounted in terms of ROTI variability on daily basis and longitudinally": is not fully clear, i.e., what is accounted for where?

**Answer**:

The sentence is paraphrased as follows:

In this article, we apply the Rankine- Hugoniot density jump relation and parameters of type II radio bursts to estimate the parameter of shock waves (shock and the Alfvén speed, the Alfvén Mach number ) of metric type II radio bursts observed by e-CALLISTO and then analyze their space weather implication in terms of the ionospheric TEC enhancements using  ROTI variability on daily basis.